# Learning Multi-Source and Robust Representations for Continual Learning

**Fei Ye[1], YongCheng Zhong[1], Qihe Liu[1],\* Adrian G. Bors[2],**
**JingLing Sun[1], RongYao Hu[1], ShiJie Zhou[1]**
[1]School of Information and Software Engineering,
University of Electronic Science and Technology of China
[2]Department of Computer Science, University of York
{feiye@uestc.edu.cn, 202422090410@std.uestc.edu.cn, qiheliu@uestc.edu.cn,
adrian.bors@york.ac.uk, jlsun@uestc.edu.cn, ryhu@uestc.edu.cn, sjzhou@uestc.edu.cn}

## Abstract

Plasticity and stability denote the ability to assimilate new tasks while preserving previously acquired knowledge, representing two important concepts in continual learning. Recent research addresses stability by leveraging pre-trained models to provide informative representations, yet the efficacy of these methods is highly reliant on the choice of the pre-trained backbone, which may not yield optimal plasticity. This paper addresses this limitation by introducing a streamlined and potent framework that orchestrates multiple different pre-trained backbones to derive semantically rich multi-source representations. We propose an innovative Multi-Scale Interaction and Dynamic Fusion (MSIDF) technique to process and selectively capture the most relevant parts of multi-source features through a series of learnable attention modules, thereby helping to learn better decision boundaries to boost performance. Furthermore, we introduce a novel Multi-Level Representation Optimization (MLRO) strategy to adaptively refine the representation networks, offering adaptive representations that enhance plasticity. To mitigate over-regularization issues, we propose a novel Adaptive Regularization Optimization (ARO) method to manage and optimize a switch vector that selectively governs the updating process of each representation layer, which promotes the new task learning. The proposed MLRO and ARO approaches are collectively optimized within a unified optimization framework to achieve an optimal trade-off between plasticity and stability. Our extensive experimental evaluations reveal that the proposed framework attains state-of-the-art performance. The source code of our algorithm is available at https://github.com/CL-Coder236/LMSRR.

## 1 Introduction

To thrive in natural environments, advanced intelligent entities must possess a robust ability to assimilate new information while retaining previously acquired critical knowledge [17]. This ability, known as continual learning (CL), is also pivotal in artificial intelligence systems, facilitating the deployment of numerous real-time applications such as autonomous driving and robotic navigation. Despite the impressive performance of contemporary deep learning models on static datasets [21], they experience substantial performance degradation in continual learning scenarios due to catastrophic forgetting [44]. This phenomenon occurs when the neural network overwrites its parameters to accommodate new task learning, leading to network forgetting.

---

\*corresponding author

39th Conference on Neural Information Processing Systems (NeurIPS 2025).

Recent research has expanded beyond the issue of catastrophic forgetting to introduce two pivotal concepts in evaluating a model's efficacy in continual learning : plasticity, which refers to the model's capacity to assimilate new tasks, and stability, which denotes its ability to retain previously acquired knowledge [28]. Most existing studies mainly focus on enhancing stability by developing several methods, which can be divided into three primary categories : Rehearsal-based techniques [10, 4], which utilize and refine a memory system to retain select historical examples; dynamic expansion-based methods [13, 24], which focus on dynamically constructing and integrating new sub-networks within a cohesive framework to accommodate new information; and regularization-based strategies [30, 42], which seek to fine-tune and adjust the model's parameters by imposing penalties on substantial alterations to critical parameters. Among these strategies, leveraging a memory system is an effective means of maintaining stability, though its efficacy diminishes significantly when the memory buffer size is constrained [60]. Conversely, dynamic expansion methods are suitable for handling extended task sequences, maintaining robust performance on historical tasks by freezing all previously trained network parameters [61]. Nonetheless, freezing the majority of the model's parameters can prevent the new task learning and thus adversely affect plasticity.

To balance stability and plasticity in continual learning, recent studies have explored pre-trained models by either extracting robust features or dynamically constructing new sub-networks based on these foundational architectures [40, 15, 43]. Nonetheless, the effectiveness of these approaches largely relies on the selection of the pre-trained backbone, which would fail to achieve optimal plasticity, particularly when confronted with novel data domains. In this study, we tackle this challenge by introducing an innovative framework named Learning Multi-Source and Robust Representations (LMSRR). This framework orchestrates several different pre-trained Vision Transformer (ViT) backbones as representation networks, delivering robust feature information to enhance performance. Specifically, we propose a novel Multi-Scale Interaction and Dynamic Fusion (MSIDF) method to proficiently amalgamate multi-source features from diverse representation networks into an augmented representation. This method captures the most important parts of the representation in response to incoming samples through several learnable attention modules, thereby enhancing plasticity. Furthermore, the proposed MSIDF approach incorporates an adaptive weighting mechanism to autonomously determine the significance of each attention module, facilitating the interaction among multi-scale features and aiding in uncovering the intricate underlying structure of the data, which further improves the model's performance.

On the other hand, numerous existing studies usually freeze the representation network to ensure stability, which inadvertently diminishes the model's capacity to learn new tasks due to the limited number of activation parameters. In this paper, we address this challenge by introducing an innovative Multi-Level Representation Optimization (MLRO) strategy. This approach incorporates a penalty term in the primary objective function, which minimizes the divergence between all previously acquired and currently activated representations, thereby maintaining stability during the new task learning. Furthermore, we propose a novel Adaptive Regularization Optimization (ARO) strategy, designed to selectively penalize parameter changes within each representation layer. Specifically, the proposed ARO approach introduces a learnable switch vector, which is dynamically optimized and continuously generates differentiable variables to selectively regulate the optimization process of each representation layer during training. Such an approach effectively relieves over-regularization issues while preserving robust plasticity. Unlike prior multi-model fusion approaches such as CoFiMA [41] and Model Soup [58], which either average independently trained models or expand architectures with task-specific modules, our LMSRR framework dynamically aggregates multiple pre-trained backbones through a unified feature-space fusion mechanism. This design enables LMSRR to adapt efficiently across tasks in continual learning scenarios without introducing additional task-specific parameters.

We conducted an extensive suite of experiments in continual learning, and the empirical findings reveal that the proposed approach attains state-of-the-art performance. The principal contributions of this research are delineated as follows :

- We propose a novel LMSRR framework to explore multi-source representations from several different pre-trained ViT backbones to boost the model's performance in continual learning.

- We propose a novel MSIDF approach to effectively integrate multi-source features into a compact and semantically rich representation, which can maintain good plasticity.

- We propose a novel MLRO approach to automatically regulate the optimization process of each representation layer, which can maintain stability during the new task learning.

- We propose a novel ARO approach to optimize a learnable switch vector that selectively penalizes the change in the parameters of each representation network, which can avoid over-regularization issues.

## 2 Related Work

**Rehearsal-based techniques** represent a widely adopted strategy for mitigating forgetting by dynamically incorporating a limited number of historical examples into the memory buffer [5, 9]. These memory samples are leveraged alongside new training instances to enhance model performance during the new task learning. Thus, the quality of the memorized samples is paramount within the rehearsal-based optimization framework [20]. Moreover, rehearsal-based approaches can be augmented through the integration of regularization techniques, with the objective of further elevating the overall efficacy of the model [2, 14, 26]. In addition, memory studies have proposed to train the generative models to implement the memory system, which can provide infinite generative replay samples [1, 47, 52, 64, 31].

**Knowledge distillation (KD) techniques** were initially developed for model compression. The fundamental concept of the KD framework involves establishing a teacher-student architecture, wherein a loss function is employed to align the predictions of the teacher and student models. This process aims to facilitate the transfer of knowledge from the complex teacher model to the simpler student model [18, 23]. KD has found extensive applications in deep learning, yielding substantial results. Given its advantageous properties and performance, KD has also been utilized to mitigate network forgetting in continual learning scenarios. The primary objective of integrating KD within continual learning is to minimize the divergence between the predictions of the student and teacher models during task learning, as outlined in Learning Without Forgetting (LWF) [37]. Moreover, rehearsal-based approaches can be synergistically combined with KD to form a unified learning framework, which has demonstrated enhanced model performance, as illustrated in [48]. Additionally, the self-KD approach has been proposed to maintain previously acquired representations, thereby alleviating network forgetting, as discussed in [9].

**Dynamic network architectures** represent a robust approach to mitigating network forgetting in continual learning [13]. Such approaches dynamically expand the network capacity to enhance the learning ability for new tasks [29, 53]. Beyond convolutional neural networks, dynamic expansion techniques have also been explored to leverage the capabilities of Vision Transformers (ViT) [15] as the foundational backbone. These methods typically create self-attention blocks combined with task-specific classifiers to adapt to new tasks [16, 59, 43]. Additionally, another investigation [46] proposes a dual learning framework that integrates a ViT with a multimodal large language model, introducing a Mises–Fisher Outlier Detection and Interaction (vMF-ODI) strategy to enhance inter-model communication. However, these methodologies often involve freezing large portions of the pre-trained backbone, which limits adaptability to complex and unseen domains. Moreover, recent architecture-based methods such as RPSNet [25] alleviate forgetting by selecting task-specific subnetworks within a shared backbone, enabling partial parameter reuse across tasks. In contrast, our LMSRR maintains a fixed architecture and performs semantic-level fusion across multiple pretrained backbones, achieving task-agnostic adaptability without subnetwork selection.

## 3 Methodology

### 3.1 Problem Statement

In continual learning (CL), models face the limitation of being unable to access the entire training dataset. The training for each task is restricted to data samples pertinent to the current task, and data from previous tasks is inaccessible. A prominent scenario in this domain is Task-Incremental Learning (TIL), where the training dataset $\mathcal{D}^s = \{(\mathbf{x}_j, \mathbf{y}_j) \mid j = 1, \cdots, N^s\}$ is divided into multiple task-specific subsets $\{\mathcal{D}_1^s, \cdots, \mathcal{D}_{C'}^s\}$, each corresponding to an individual task $\mathcal{T}_j$. During the learning of a specific task $\mathcal{T}j$, the model is confined to data samples from the relevant training subset $\mathcal{D}_j^s$, while all prior subsets $\{\mathcal{D}_1^s, \cdots, \mathcal{D}_{C'}^s\}$ remain inaccessible. In each task, the model learns to discriminate

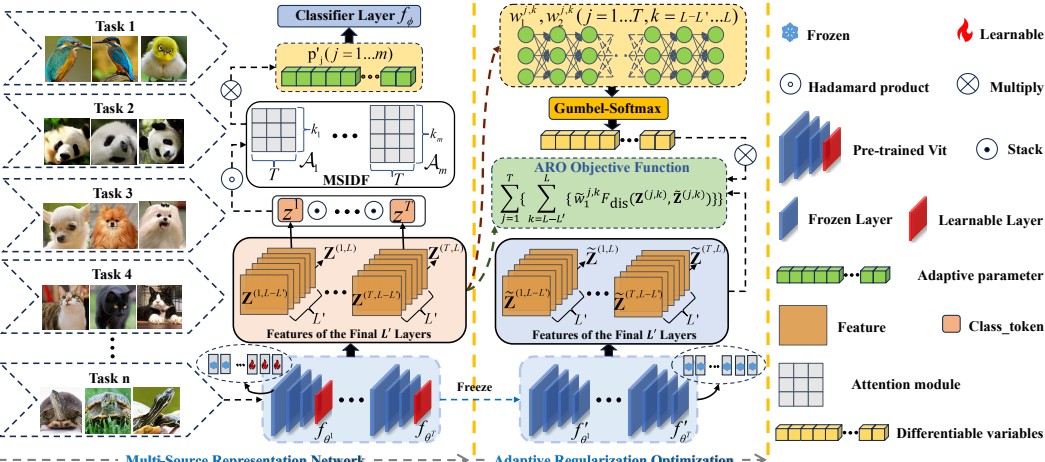

Figure 1: The overall framework of the LMSRR. During training, only the last $L'$ layers of each ViT backbone are trainable, with the rest frozen. Data samples are processed by these ViT backbones to extract feature outputs, which are subsequently stacked. The stacked features are integrated through the proposed MSIDF module before being passed to a fully connected classifier for final prediction. In addition, the proposed MLRO approach optimizes the representation networks by penalizing shifts in the parameters, which can ensure the preservation of all previously learned information. Furthermore, we introduce a novel ARO approach to adaptively regulate the optimization process of the representation networks, which can relieve over-regularization issues.

among classes within that task, and the task identifier is provided during both training and evaluation, allowing the model to use task-specific output heads or parameters when necessary.

The goal of a model in continual learning is to progressively optimize the parameters as new task data is introduced, minimizing the overall training loss across all tasks. Specifically, the model aims to find the optimal set of parameters $\theta^\star$ from the parameter space $\tilde{\Theta}$, such that the loss function is minimized over all training samples from each task. This problem can be formalized as the following optimization problem :

$$\theta^\star = \underset{\theta \in \tilde{\Theta}}{\arg\min} \frac{1}{j} \sum_{k=1}^{j} \left\{ \frac{1}{N_j^s} \sum_{c=1}^{N_j^s} \left\{ \mathcal{L}\left(\mathbf{y}_c, f_\theta(\mathbf{x}_c)\right) \right\} \right\}, \tag{1}$$

where $\theta^\star$ represents the optimal model parameters, and $\mathcal{L}(\cdot, \cdot)$ is the loss function, which is commonly implemented as the cross-entropy loss to measure the discrepancy between model predictions and true labels. The function $f_\theta(\cdot) \colon \mathcal{X} \to \mathcal{Y}$ represents the classifier with parameter set $\theta$, which maps input samples $\mathbf{x}_c \in \mathcal{X}$ to their predicted labels $\mathbf{y}_c \in \mathcal{Y}$, where $\mathcal{X}$ and $\mathcal{Y}$ denote the data and class label space, respectively. $N_j^s$ is the total number of samples in the training subset $\mathcal{D}_j^s$. Due to the inaccessibility of historical examples in continual learning, many studies have implemented the goal of Eq. (1) by proposing to employ a memory system to preserve historical examples.

After completing the learning of all tasks $\{\mathcal{T}_1, \cdots, \mathcal{T}_N\}$, the model's performance is evaluated using all test datasets $\{D_1^t, \cdots, D_N^t\}$. This evaluation not only considers the model's performance on the current task but also examines its performance on previous tasks, providing a comprehensive assessment of the model's ability to adapt to a continuously changing data distribution.

## 3.2 Multi-Source Representation Network

Acquiring robust and semantically enriched representations can markedly enhance model performance across diverse applications [6]. Numerous studies have leveraged pre-trained neural networks to deliver potent and resilient representations, with the objective of augmenting performance in continual learning [45, 65]. Nonetheless, these approaches need to carefully select an appropriate pre-trained backbone, which may not achieve optimal plasticity when confronted with novel data domains. In this study, we propose an innovative framework to manage and optimize several different pre-trained Vision Transformers (ViTs) as foundational representation networks, thereby providing robust and

semantically enriched representations for continual learning. Let $f_{\theta^i} : \mathcal{X} \to \mathcal{Z}$ denote the $i$-th pre-trained ViT backbone, which processes the image $\mathbf{x} \in \mathcal{X}$ as input and outputs a feature vector $\mathbf{z} \in \mathcal{Z}$, where $i = 1, \cdots, T$ and $T$ signifies the total number of ViT backbones. Here, $\mathcal{Z} \in \mathbf{R}^{d_z}$ and $\mathcal{X} \in \mathbf{R}^{d_x}$ represent the feature and data spaces, respectively, with $d_z$ and $d_x$ as their respective dimensions.

Integrating the output features from various representation networks, each containing distinct intrinsic properties, can yield a rich diversity of representational information. A straightforward and effective method involves consolidating multi-source features into a unified representation for a specific data point $\mathbf{x}_s$, as described by :

$$\mathbf{z}'_s = f_{\theta^1}(\mathbf{x}_s) \otimes \cdots \otimes f_{\theta^T}(\mathbf{x}_s) , \tag{2}$$

where $\otimes$ signifies the fusion of several feature vectors into an expanded dimensional space. Utilizing the enhanced representation $\mathbf{z}'_s$, we can dynamically create a new expert to learn a decision boundary for a specific task, aiming to implement the prediction process. Specifically, the expert is implemented using a linear classifier $f_\phi : \mathcal{Z}^a \to \mathcal{Y}$, which receives an augmented representation and returns a prediction, expressed as :

$$\mathbf{y}'_s = f_\phi(f_{\theta^1}(\mathbf{x}_s) \otimes \cdots \otimes f_{\theta^T}(\mathbf{x}_s)) , \tag{3}$$

where $\mathbf{y}'_s = \{y'_{1,s}, \cdots, y'_{C,s}\}$ denotes the predicted probabilities, with $C$ signifying the total number of categories. $\mathcal{Z}^a \in \mathbf{R}^{d_{z^a}}$ denotes the $d_{z^a}$-dimensional feature space associated with the augmented representation $\mathbf{z}'_s$, while $\mathcal{Y} \in \mathbf{R}^{d_y}$ represents the $d_y$-dimensional prediction space. Unlike model-averaging or ensemble-based approaches that combine multiple independently trained models, our framework performs feature-space fusion of several pre-trained ViT backbones within a unified continual learning setup, maintaining a fixed inference path without parameter growth.

## 3.3 Multi-Scale Interaction and Dynamic Fusion

The augmented representations formulated in Eq. (2) assume an equal contribution from each representation network towards the learning of a new task. However, this approach does not fully exploit the representational capacity. Moreover, simply combining these multi-source features can cause redundancy in the representational information, resulting in performance degradation. In this research, we tackle these issues by introducing an innovative MSIDF mechanism that autonomously filters out redundant information while preserving essential feature components. Specifically, for a given input $\mathbf{x}_s$, the proposed MSIDF mechanism initially constructs an augmented representation by :

$$\tilde{\mathbf{z}}_s = f_{\theta^1}(\mathbf{x}_s) \bullet \cdots \bullet f_{\theta^T}(\mathbf{x}_s) , \tag{4}$$

where $\bullet$ signifies the operation that stacks multiple vectors $\{f_{\theta^1}(\mathbf{x}_s), \cdots, f_{\theta^T}(\mathbf{x}_s)\}$ into a matrix $\tilde{\mathbf{z}}_s \in \mathbf{R}^{T \times d_z}$. Subsequently, the proposed MSIDF framework introduces a set of adaptive attention modules $\{\mathcal{A}_1, \cdots, \mathcal{A}_m\}$, where each attention module $\mathcal{A}_j$ is characterized by a learnable matrix $\mathbf{W}^j \in \mathbf{R}^{k_j \times T}$ with a window size $k_j$, designed to discern the most pertinent feature components. The process of using a specific attention module (the $j$-th module) to the representation matrix $\tilde{\mathbf{z}}_c$ is articulated as follows :

$$F_t(\tilde{\mathbf{z}}_s, i) = \mathbf{W}^j \circ \tilde{\mathbf{z}}_s[:][i : i + k_j] , \tag{5}$$

where $\circ$ denotes the Hadamard product and $\tilde{\mathbf{z}}_s[:][i : i + k_j]$ denotes a matrix starting from the row $i$ and ending at the row $i + k_j$. By using Eq. (5), we can form a processed representation by :

$$\mathbf{Z}^j_s = F_t(\tilde{\mathbf{z}}_s, 0) \otimes, \cdots, \otimes F_t(\tilde{\mathbf{z}}_s, d_z - k_j + 1) , \tag{6}$$

where $\mathbf{Z}^j_s$ denotes a representation refined through the $j$-th attention module. For attention modules with varying window sizes, we utilize symmetric padding techniques to ensure that the dimensions of the representations processed by each attention module are consistent with those of other attention modules. Furthermore, to facilitate the cooperative optimization of these attention modules, the proposed MSIDF mechanism introduces a trainable adaptive parameter $p_j$ to ascertain the significance of each $\mathcal{A}_j$ during the training phase. To prevent numerical overflow, we normalize each trainable adaptive parameter $p_j$ by :

$$p'_j = \exp(p_j) / \sum\nolimits_{c=1}^{m} \exp(p_c) . \tag{7}$$

By using the adaptive weights, all processed representations $\{\mathbf{Z}^1_s, \cdots, \mathbf{Z}^m_s\}$ are integral by :

$$\mathbf{Z}_s = \sum\nolimits_{j=1}^{m} \left\{ p'_j \mathbf{Z}^j_s \right\} , \tag{8}$$

where $\mathbf{Z}_s$ denotes the ultimate augmented representation, which is fed into a linear classifier for prediction. In contrast to Eq. (2), Eq. (8) can provide a more concise and informative representation, maintaining a constant feature dimension even as the number of representation networks increases.

## 3.4 Multi-Level Representation Optimization

Refining the parameters of representation networks can facilitate the acquisition of new tasks, thereby enhancing their plasticity. Nevertheless, optimizing the entire parameter set of the model is computationally intensive due to the substantial number of hidden layers and nodes within each representation network. Recent research has shown that high-level representations from large-scale pre-trained neural networks provide semantically rich information, which enhances model performance in downstream tasks [38, 62]. Based on these empirical insights, we propose optimizing only the last $L'$ layers to mitigate computational demands. To ensure stability in continual learning, we introduce an innovative MLRO method, which regulates the representation updating behaviour during the optimization process. Specifically, let $f'_{\theta^j}$ denote a representation network trained on the preceding task ($T_{i-1}$) and kept static during the learning of a new task ($T_i$), while $f_{\theta^j}$ is the active representation network during the new task learning ($T_i$), where $j = 1, \cdots, T$. Each representation network $f_{\theta^j}$ consists of $L'$ trainable feature layers, represented as $\{f_{\theta^j_{L-L'}}, \cdots, f_{\theta^j_L}\}$, where each $f_{\theta^j_c} : \mathcal{Z}^{c-1} \to \mathcal{Z}^c$ processes the representation over the feature space $\mathcal{Z}^{c-1}$ extracted by $f_{\theta^j_{c-1}}$ and outputs the representation over the feature space $\mathcal{Z}^c$. A representation extracted by a specific feature layer of a representation network is articulated as follows :

$$F_{\mathrm{f}}(f_{\theta^j}, \mathbf{x}, k) = \begin{cases} f_{\theta^j_1}(\mathbf{x}) & k = 1 \\ f_{\theta^j_2}(f_{\theta^j_1}(\mathbf{x})) & k = 2 \\ f_{\theta^j_k}(\cdots f_{\theta^j_2}(f_{\theta^j_1}(\mathbf{x}))) & 3 \le k \le L. \end{cases} \tag{9}$$

For a given data batch $\mathbf{X} = \{\mathbf{x}_1, \cdots, \mathbf{x}_b\}$ at the $i$-th task learning, we extract the representations using the $j$-th active representation network, expressed as :

$$F_z(\mathbf{X}, f_{\theta^j}, k) = \Big\{ \mathbf{z}_s \,|\, \mathbf{z}_s = F_{\mathrm{f}}(f_{\theta^j}, \mathbf{x}_s, k), s = 1, \cdots, b \Big\}, \tag{10}$$

where $b$ denotes the batch size. We can obtain a collection of feature vectors $\{\mathbf{Z}^{(j,L-L')}, \cdots, \mathbf{Z}^{(j,L)}\}$ by leveraging the last $L'$ active feature layers of the $j$-th backbone $f_{\theta^j}$, where each $\mathbf{Z}^{(j,k)}$ is computed using $F_z(\mathbf{X}, f_{\theta^j}, k)$. Similarly, we utilize each frozen representation network $f'_{\theta^j}$ to extract a set of previously acquired feature vectors $\{\tilde{\mathbf{Z}}^{(j,L-L')}, \cdots, \tilde{\mathbf{Z}}^{(j,L)}\}$ using Eq. (10), with $\tilde{\mathbf{Z}}^{(j,k)} = F_z(\mathbf{X}, f'_{\theta^j}, k)$. The proposed MLRO approach incorporates a regularization loss component aimed at minimizing the divergence between the previously acquired and currently active representations, formulated as follows :

$$F_{\mathrm{re}}(\mathbf{X}) = \sum_{j=1}^{T} \Big\{ \sum_{k=L-L'}^{L} \big\{ F_{\mathrm{dis}}(\mathbf{Z}^{(j,k)}, \tilde{\mathbf{Z}}^{(j,k)}) \big\} \Big\}, \tag{11}$$

where $F_{\mathrm{dis}}(\cdot, \cdot)$ represents a generic distance metric used to quantify the divergence between two sets of feature vectors. We opt for the L2 distance due to its computational efficiency and straightforward implementation. Furthermore, to address the shifts in the representations of historical examples, we incorporate a memory buffer $\mathcal{M}$ designed to store and maintain numerous past instances. As the primary focus of this paper is on optimizing representation strategies rather than the memory system, we consider employing a simple reservoir sampling method [54] for memory updates, ensuring computational efficiency.

## 3.5 Adaptive Regularization Optimization

The representation optimization process, as delineated in Eq. (11), presupposes uniform regularization intensity across all representation layers during training, which may not yield optimal plasticity. This paper tackles this limitation by introducing an innovative ARO method that selectively constrains parameter alterations in each representation layer throughout the optimization process. Specifically, the proposed ARO method incorporates a trainable switch vector $\{w_1^{j,k}, w_2^{j,k}\}$ for the $k$-th trainable feature layer within the $j$-th representation network, where $w_1^{j,k}$ and $w_2^{j,k}$ represent the probabilities

of activation and deactivation of the $k$-th representation layer, respectively. A straightforward method to determine the penalization of changes involves converting the switch vector to one-hot encoding; however, this approach lacks differentiability. To overcome this challenge, we propose utilizing the Gumbel-Softmax distribution [19] to produce differentiable variables, expressed as :

$$\tilde{w}_1^{j,k} = \frac{\exp((\log(w_1^{j,k}) + g_1)/\tau)}{\sum_{t=1}^{2}\{\exp((\log(w_t^{j,k}) + g_t)/\tau)\}} \,, \tag{12}$$

where $g_t$ is drawn from Gumbel(0,1) and $\tilde{w}_1^{j,k}$ is the differentiable approximation of $w_1^{j,k}$. $\tau$ represents a temperature parameter and a large $\tau$ encourages samples from the Gumbel Softmax distribution to become one-hot representations. Using differentiable category variables defined in Eq. (12) can derive a new regularization loss function :

$$F_{\mathrm{A}}(\mathbf{X}) = \sum_{j=1}^{T}\left\{\sum_{k=L-L'}^{L}\left\{\tilde{w}_1^{j,k}F_{\mathrm{dis}}(\mathbf{Z}^{(j,k)}, \tilde{\mathbf{Z}}^{(j,k)})\right\}\right\}, \tag{13}$$

Compared to Eq. (11), the regularization loss term defined in Eq. (13) can selectively penalize the changes in the parameters of each representation layer, which can relieve over-regularization issues and enhance plasticity.

## 3.6 The Optimization Framework

---

**Algorithm 1** Training procedure of LMSRR

---

**Require:** Number of tasks $N$, dataset $\{\mathcal{D}_1^S, \ldots, \mathcal{D}_N^S\}$, training iterations per task $n$
**Ensure:** Trained parameters of $\{f_{\theta^1}, \ldots, f_{\theta^T}\}$ and classifier $f_\phi$
  **for** $i = 1$ to $N$ **do**
    **for** $j = 1$ to $n$ **do**
      **Step 1: Multi-source feature construction**
      Sample a minibatch $\mathbf{X} = \{\mathbf{x}_1, \ldots, \mathbf{x}_b\}$ from $\mathcal{D}_i^S$
      Compute multi-source backbone outputs $\{f_{\theta^1}(\mathbf{x}_s), \ldots, f_{\theta^T}(\mathbf{x}_s)\}$
      Construct stacked representations $\tilde{\mathbf{z}}_s$ using Eq. (4)
      Obtain fused representations $\mathbf{Z}_s$ through the MSIDF module using Eq. (8)

      **Step 2: Representation-level regularization**
      Obtain active representations $\mathbf{Z}^{(j,k)}$ from the last $L'$ layers using Eq. (10)
      Obtain frozen references $\tilde{\mathbf{Z}}^{(j,k)}$ using Eq. (10)
      Compute adaptive regularization $F_{\mathrm{A}}(\mathbf{X})$ using Eq. (13)

      **Step 3: Parameter update**
      Compute total loss $\mathcal{L}_{\mathrm{loss}}$ using Eq. (14)
      Update the model's parameters
    **end for**
  **end for**

---

The proposed framework involves $T$ representation networks $\{f_{\theta^1}, \cdots, f_{\theta^T}\}$ and a linear classifier $f_\phi$. In order to update the parameters of these modules, we introduce a unified objective function at the $i$-th task learning ($T_i$), defined as :

$$\begin{aligned}\mathcal{L}_{\mathrm{loss}} = &\mathbb{E}_{(\mathbf{X},\mathbf{Y})\sim\mathbb{P}_{\mathcal{D}_i^s\otimes\mathcal{M}}}\Big[\sum_{k=1}^{b}\{F_{\mathrm{ce}}(\mathbf{y}, f_\phi(\mathbf{Z}_k))\}\Big]\\ &+ \lambda\Big(\mathbb{E}_{(\mathbf{X},\mathbf{Y})\sim\mathbb{P}_{\mathcal{M}}}\left[F_{\mathrm{A}}(\mathbf{X})\right]\\ &+ \mathbb{E}_{(\mathbf{X},\mathbf{Y})\sim\mathbb{P}_{\mathcal{D}_i^s}}\left[F_{\mathrm{A}}(\mathbf{X})\right]\Big),\end{aligned} \tag{14}$$

where $\mathbb{P}_{\mathcal{D}_i^s}$ and $\mathbb{P}_{\mathcal{D}_i^s}$ denote the distribution of the dataset $\mathcal{D}_i^s$ and the memory buffer $\mathcal{M}$, respectively. $\mathbb{P}_{\mathcal{D}_i^s\otimes\mathcal{M}}$ denotes the distribution of the combined dataset $\mathcal{D}_i^s$ and $\mathcal{M}$. $F_{\mathrm{ce}}(\cdot, \cdot)$ is the cross-entropy function and $\lambda$ is a hyperparameter that controls the effects of the regularization term during the optimization process. We provide the detailed learning process of the proposed framework in Fig. 1 while the detailed pseudocode is provided in **Algorithm 1** which consists of three steps :

**Step 1. Form augmented representations :** For a given data batch $\mathbf{X} = \{\mathbf{x}_1, \cdots, \mathbf{x}_b\}$, we can obtain fused representations $\mathbf{Z}_s$ through the MSIDF module using Eq. (8).

**Step 2. Adaptive representation optimization :** For a given data batch $\mathbf{X} = \{\mathbf{x}_1, \cdots, \mathbf{x}_b\}$, we can get all active representations $\{\mathbf{Z}^{(1,L)}, \cdots, \mathbf{Z}^{(T,L)}\}$ as well as all previously learned representations $\{\tilde{\mathbf{Z}}^{(1,L)}, \cdots, \tilde{\mathbf{Z}}^{(T,L)}\}$ using Eq. (10). The regularization term is calculated using Eq. (13).

**Step 3. Optimizing the model :** We update all model parameters $\{\phi, \mathbf{W}^1, \cdots, \mathbf{W}^m\}$ using Eq. (14). In addition, we also update the adaptive parameters $\{p_1, \cdots, p_m\}$ as well as the parameters $\{w_1^{1,L-L'}, w_2^{1,L-L'}, \cdots, w_1^{T,L}, w_2^{T,L}\}$ of the proposed ARO approach using Eq. (14).

Table 1: The classification accuracy on standard datasets is presented as the average over three runs. "Average" denotes the average accuracy across all tasks, while "Last" indicates the accuracy of the final task. The "-" in the table signifies that experiments could not be conducted due to compatibility issues or intractable training time problems.

| Buffer | Method | CIFAR-10 | | Tiny ImageNet | | R-MNIST |
| --- | --- | --- | --- | --- | --- | --- |
| | | Average | Last | Average | Last | Domain-IL |
| - | EWC [51] | 68.29±3.92 | **97.07±0.74** | 19.20±0.31 | 75.15±3.18 | 77.35±5.77 |
| | SI [63] | 68.05±5.91 | 94.18±0.88 | 36.32±0.13 | 65.80±3.25 | 71.91±5.83 |
| | LwF [37] | 63.29±2.35 | 96.75±0.35 | 15.85±0.58 | 77.95±3.60 | - |
| | PNN [50] | 95.13±0.72 | 96.63±0.10 | 67.84±0.29 | 69.03±1.01 | |
| | DAP [27] | **97.13±2.06** | 96.05±3.39 | **92.49±0.60** | **94.95±1.20** | **88.58±2.53** |
| 200 | ER [49] | 91.19±0.94 | 97.50±0.35 | 38.17±2.00 | 79.40±0.28 | 85.01±1.90 |
| | GEM [39] | 90.44±0.94 | 96.60±0.35 | - | - | 80.80±1.15 |
| | A-GEM [12] | 83.88±1.49 | 97.90±0.07 | 22.77±0.03 | 78.65±3.32 | 81.91±0.76 |
| | iCaRL [48] | 88.99±2.13 | 97.07±0.32 | 28.19±1.47 | 47.45±0.78 | - |
| | FDR [7] | 91.01±0.68 | 97.78±0.24 | 40.36±0.68 | 81.40±0.70 | 85.22±3.35 |
| | GSS [3] | 88.80±2.89 | 97.42±0.24 | - | - | 79.50±0.41 |
| | HAL [11] | 82.51±3.20 | 94.60±0.14 | - | - | 84.02±0.98 |
| | DER [8] | 91.40±0.92 | 97.80±0.28 | 40.22±0.67 | 79.15±0.21 | 90.04±2.61 |
| | DER++ [8] | 91.92±0.60 | 97.72±0.38 | 40.87±1.16 | 78.35±0.49 | 90.43±1.87 |
| | DER++(re) [56] | 92.01±3.03 | 97.65±3.03 | 47.61±8.87 | 81.40±1.41 | 91.64±2.26 |
| | **Ours** | **98.85±0.05** | **99.35±0.21** | **92.08±0.31** | **96.00±0.01** | **94.20±1.24** |
| 500 | ER [49] | 93.61±0.27 | 97.15±0.28 | 48.64±0.46 | 80.80±1.69 | 88.91±1.44 |
| | GEM [39] | 92.16±0.69 | 96.63±0.17 | - | - | 81.15±1.98 |
| | A-GEM [12] | 89.48±1.45 | 97.40±0.78 | 25.33±0.49 | 81.00±0.42 | 80.31±6.29 |
| | iCaRL [48] | 88.22±2.62 | 96.57±0.10 | 31.55±3.27 | 50.65±1.20 | - |
| | FDR [7] | 93.29±0.59 | 97.32±0.24 | 49.88±0.71 | 81.10±0.56 | 89.67±1.63 |
| | GSS [3] | 91.02±1.57 | 96.97±0.24 | - | - | 81.58±0.58 |
| | HAL [11] | 84.54±2.36 | 94.22±0.60 | - | - | 85.00±0.96 |
| | DER [8] | 93.40±0.39 | 97.90±0.28 | 51.78±0.88 | 79.30±1.13 | 92.24±1.12 |
| | DER++ [8] | 93.88±0.50 | 98.10±0.01 | 51.91±0.68 | 76.20±5.23 | 92.77±1.05 |
| | DER++(re) [56] | 93.06±0.38 | 97.75±0.38 | 54.06±0.79 | 79.65±1.34 | 93.28±0.75 |
| | **Ours** | **99.15±0.05** | **99.48±0.04** | **92.75±0.32** | **96.23±0.40** | **96.97±1.58** |
| 1000 | ER [49] | 95.34±0.16 | 97.67±0.67 | 55.92±0.90 | 80.30±0.82 | 90.42±1.07 |
| | GEM [39] | 93.67±0.32 | 97.37±0.17 | - | - | 81.15±1.98 |
| | A-GEM [12] | 85.61±2.01 | 97.45±0.42 | 24.29±1.28 | 79.65±2.19 | 81.30±5.33 |
| | iCaRL [48] | 91.40±1.06 | 96.85±0.35 | 63.87±0.25 | 54.00±2.82 | - |
| | FDR [7] | 94.02±0.64 | 97.60±0.56 | 56.05±0.71 | 80.25±0.49 | 91.68±1.01 |
| | GSS [3] | 91.79±2.16 | 96.10±1.70 | - | - | 82.25±2.42 |
| | HAL [11] | 87.33±1.46 | 92.27±3.21 | - | - | 89.33±2.01 |
| | DER [8] | 92.33±0.61 | 97.72±0.07 | 56.62±1.13 | 78.50±0.42 | 93.13±0.28 |
| | DER++ [8] | 94.99±0.26 | 97.94±0.08 | 58.05±0.52 | 79.95±0.35 | 93.82±0.39 |
| | DER++(re) [56] | 93.66±1.00 | 97.40±0.01 | 61.91±1.15 | 80.45±3.18 | 93.37±0.58 |
| | **Ours** | **99.21±0.06** | **99.43±0.03** | **93.24±0.24** | **96.10±0.57** | **97.05±0.04** |

# 4 Experiment

## 4.1 Experimental setting

**Datasets.** we conducted extensive experiments on seven different datasets, including CIFAR-10 [33], TinyImageNet [35], MNIST [36], CIFAR-100 [34], CUB-200 [55], ImageNet-R [22], and Cars196 [32]. We provide the detailed experiment setting in **Appendix A** from Supplementary Material (SM).

## 4.2 Results on Standard Datasets

In this section, we compare the proposed approach with several baselines on the standard datasets, including CIFAR-10, Tiny ImageNet and R-MNIST, under memory buffer sizes of 200, 500, and 1000. The empirical results are reported in Tab. 1 . These results show that LMSRR significantly outperforms the other baselines in terms of classification accuracy. This highlights LMSRR's ability to effectively retain previously acquired knowledge as the number of tasks increases, demonstrating its remarkable plasticity and resistance to catastrophic forgetting.

Previous CL methods, such as EWC, SI, and LwF, have lower average accuracy. The reason behind this result is that regularization-based methods typically degrade when the new task contains abundant different information with respect to prior tasks. PNN, as a dynamic expansion model, still struggles with scalability when dealing with long sequences of tasks, which significantly reduces its perfor-

Table 2: The classification results of various models on complex datasets, with a memory buffer size of 500, calculated as the average results of three independent runs.

| Method | CIFAR-100 | | CUB-200 | | Imagenet-R | | Cars196 | |
|---|---|---|---|---|---|---|---|---|
| | Average | Last | Average | Last | Average | Last | Average | Last |
| ER [49] | 73.37±0.43 | 93.35±1.34 | 30.57±4.81 | 35.57±14.86 | 24.85±4.06 | 45.85±0.01 | 30.52±4.4 | 54.32±5.07 |
| A-GEM [12] | 48.06±0.57 | 92.80±0.32 | 13.22±0.31 | 42.18±0.01 | 16.87±2.65 | 47.56±12.31 | 8.07±0.15 | 16.45±7.41 |
| FDR [7] | 76.29±1.44 | 93.60±1.34 | 23.94±0.07 | 45.58±0.19 | 15.74±3.69 | 42.14±10.75 | 31.41±1.30 | 58.36±1.17 |
| GSS [3] | 57.50±1.93 | 92.80±2.98 | 27.04±0.28 | 42.01±0.08 | 17.83±0.88 | 33.44±6.75 | 34.67±2.27 | 56.80±4.15 |
| DER [8] | 74.93±1.06 | 93.25±0.35 | 26.19±2.07 | 51.79±1.08 | 18.26±1.67 | 25.26±0.47 | 39.75±0.36 | 68.02±5.20 |
| DER++ [8] | 75.64±0.60 | 92.60±0.14 | 33.40±1.48 | 49.83±1.63 | 22.87±5.83 | 43.10±10.51 | 35.39±3.38 | 60.56±8.45 |
| DER++refresh [56] | 77.71±0.85 | 93.40±1.13 | 35.77±3.20 | 50.85±0.47 | 23.74±3.03 | 31.00±0.01 | 33.94±2.46 | 60.29±4.73 |
| CoFiMA [41] | 94.21±0.47 | 96.13±0.59 | **90.66±0.76** | 92.54±0.28 | 83.76±0.53 | 85.86±0.58 | 87.28±0.54 | 90.33±0.45 |
| DAP [27] | 90.11±0.33 | 92.30±2.12 | 71.83±1.44 | 72.23±2.85 | 83.22±1.25 | 84.61±2.85 | 39.79±1.85 | 65.35±2.21 |
| L2P [57] | 95.36±0.12 | 96.80±0.14 | 86.30±0.21 | 90.81±0.24 | **86.01±0.30** | 87.50±0.90 | 79.55±0.86 | 84.45±0.12 |
| **Ours** | **95.76±0.08** | **98.70±0.37** | 88.91±0.64 | **94.31±0.12** | 84.35±0.52 | **88.43±0.15** | **90.14±0.06** | **95.32±0.39** |

mance. Experience replay-based methods, such as GEM, GSS, DER, DER++, and DER++refresh, experience noticeable performance drops when the memory buffer is limited. This indicates that these methods struggle to capture critical informative samples when the memory buffer is constrained. Notably, our model maintains excellent performance even with a small buffer size, further proving its adaptability and effectiveness across various continual learning scenarios.

### 4.3 Results on Complex Datasets

We evaluate our method against various baselines on complex datasets, and report the average and last accuracy in Tab. 2. Replay-based methods such as ER, DER, and GSS show clear performance degradation on complex datasets, reflecting their limited ability to capture fine-grained visual semantics when constrained by a fixed memory buffer. Although DAP and L2P leverage prompt-based mechanisms to mitigate representation drift and achieve better adaptation, their performance still relies heavily on the alignment between the pre-trained backbone and the target domain. For example, L2P performs well on ImageNet-R but struggles on Cars196, where the distribution gap from pre-training data is large.

CoFiMA, which employs a multi-model ensemble strategy through fixed-weight logit-level integration and introduces a new adapter for each task, shows strong results on CUB-200, benefiting from its ability to preserve task-specific knowledge. However, its design leads to parameter growth and task-dependent routing during inference, which limits scalability. In contrast, LMSRR attains consistently superior or comparable performance across all datasets within a unified architecture, achieving the highest results on CIFAR-100 and Cars196.

### 4.4 Ablation Study

In this section, we perform a full ablation study experiment to investigate the performance of the LMSRR with different configurations. More ablation study results are provided in **Appendix B** from SM.

**Backbone.** To ensure a fair comparison, we adopted the same multiple pre-trained ViT models as our method's backbone for other SOTA methods that do not involve modifications to the backbone network structure. In these methods, each pre-trained ViT model is only allowed to update the parameters of the last three feature layers. The feature representations extracted by each pre-trained ViT are concatenated and then fed into a linear classifier to obtain the final output. Fig. 2(a) shows the average accuracy of our method and SOTA models on the ImageNet-R dataset under different memory buffer configurations. The results indicate that our method consistently achieved the highest accuracy across various buffer sizes and significantly outperformed other models.

**Forgetting rates.** Fig. 2(b) presents the forgetting curves of our method and other methods on the ImageNet-R dataset. The results show that some SOTA models exhibit significant forgetting, especially static models like ER and DER, whose performance drops notably as the number of tasks increases. In contrast, our method maintains stable and superior performance, achieving the lowest forgetting rate. This is attributed to our MLRO technique, which continuously adjusts the representation optimization process over time, effectively mitigating network forgetting.

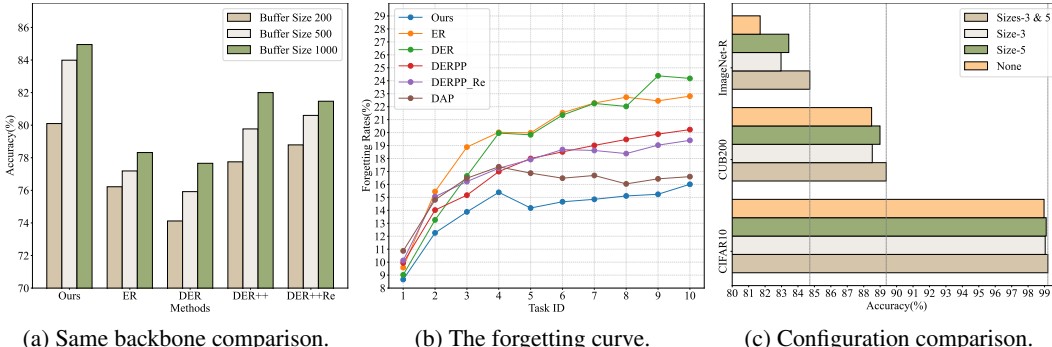

|  (a) Same backbone comparison. | (b) The forgetting curve. | (c) Configuration comparison. |

Figure 2: (a) Comparison of performance of various models with varying buffer sizes on ImageNet-R, where each model uses the same backbone. (b) Comparison of forgetting curves of the proposed approach with other benchmark methods on ImageNet-R. (c) Performance variations of the proposed MSIDF method under different configurations.

**Different configurations.** The MSIDF is driven by multiple attention modules of varying sizes, which can impact model performance based on their dimensions and quantity. To evaluate the MSIDF mechanism, we test the following four configurations across multiple datasets: MSIDF with two attention modules of different sizes-3 & 5; MSIDF with only a size-3 attention module; MSIDF with only a size-5 attention module; and a baseline model without the MSIDF mechanism. The experimental results, as shown in Fig. 2(c), indicate that the MSIDF with two differently sized attention modules achieved the highest classification accuracy, and models using MSIDF outperformed the baseline model without this mechanism. These findings highlight the significance of MSIDF in enhancing overall model performance by effectively capturing more critical feature information through attention modules of diverse sizes.

# 5 Conclusion and Limitation

This work introduced LMSRR, a framework that leverages multiple pre-trained ViT backbones to obtain diverse and complementary representations, employs MSIDF for multi-scale feature interaction, and incorporates MLRO and ARO to balance plasticity and stability through representation-level regularization and adaptive layer-wise constraints. Extensive experiments across seven datasets demonstrate that LMSRR consistently improves accuracy and mitigates forgetting under various memory budgets. A limitation of the current study is that LMSRR relies on a fixed collection of pre-trained backbones and has not yet explored scalability to broader backbone families or multimodal environments. Future work will investigate more flexible backbone integration strategies and extend the framework to more dynamic and open-world continual learning scenarios.

# 6 Acknowledgements

This work was supported by the National Natural Science Foundation of China (Grant Nos. 62506067), the Sichuan Provincial Natural Science Foundation Project (Grant No. 2025ZNS-FSC0510), and the Fundamental Research Funds for the Central Universities (Grant Nos. ZYGX2025XJ024 and ZYGX2025XJ025).

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
