# OpenReview forum: "Learning Multi-Source and Robust Representations for Continual Learning"
_NeurIPS.cc/2025/Conference — NeurIPS 2025 poster_

### Official Review · Reviewer_3Jvb · 2025-06-26

**Clarity:** 3
**Significance:** 2
**Originality:** 2
**Rating:** 4
**Confidence:** 4

**Summary:**

In this paper, the authors propose a new method for class-incremental learning problem. Instead of a single pre-trained model, the authors propose to orchestrate multiple different pre-trained backbones to derive semantically rich multi-source representations. As for optimization, the authors also propose Multi-Level Representation Optimization strategy to adaptively refine the backbones. The experiments show the effectiveness of the proposed method.

**Questions:**

Please refer to the weaknesses.

**Ethical Concerns:**

["NO or VERY MINOR ethics concerns only"]

**Final Justification:**

Most of my problems have been solved by the authors' response. I have raised my rating to recommend acceptance.

**Limitations:**

Please refer to the weaknesses.

**Paper Formatting Concerns:**

There is no formatting problem.

**Quality:**

2

**Strengths And Weaknesses:**

Strength:
1. The integration of leveraging the knowledge of pre-trained models and selectively optimize them shows the great improvement over class-incremental learning.
2. The writing and mathematical explanation makes the idea easy to understand

Weakness:
1. The performance comparison is unfair. Most of the methods are proposed before 2021 and the most recent work DAP is proposed in 2023. However, there has been some improvements in class-incremental learning in recent years so that the authors should compare with more SOTA. Second, the compared methods are mostly replay-based or regularization-based but the proposed method is more likely the architecture-based that is naturally more effective for catastrophic forgetting. The authors should compare with some similar methods like [1] and [2] that leverages a combination of small models and optimize part of the representations. Third, the proposed method is based on pre-trained models that already have good representations while the compared methods are not pre-trained. Therefore, the authors should compare with methods with pre-trained knowledge, such as [3], to show their effectiveness.
2. I wonder how the performance changes when the number of experts changes.
3. I wonder whether the proposed method is more effective than compared methods when the first task contains more classes.

[1] Random path selection for continual learning

[2] Overcoming catastrophic forgetting with hard attention to the task

[3] Learning to prompt for continual learning

---

> ### Author Rebuttal · Authors · 2025-07-31
>
> **Q1:Fairness of performance comparison, architectural differences, and pre-training strategy**
>
> We sincerely thank the reviewer for their professional comments regarding the categorization of our method and the fairness of our experimental comparisons. We provide a unified response to the three sub-issues raised:
>
> **1. Key Distinction from Architecture-Expansion Methods**
>
> Although LMSRR utilizes multiple pre-trained ViT models as feature extractors, our method does not expand model structures per task (e.g., by adding branches or task-specific heads). Instead, our goal is to construct a semantically aligned multi-source representation space within a unified architecture.
>
> Specifically, feature representations from multiple pre-trained models are fused via our proposed MSIDF (Multi-Source Interactive Dynamic Fusion) module. This fused representation is then progressively optimized through MLRO (Multi-Level Representation Optimization) and ARO (Auxiliary Representation Optimization) modules. This design maintains feature continuity and robustness across tasks, and is fundamentally different from task-isolated routing strategies like those used in RPSNet [1] and HAT [2], which suffer from parameter growth as the number of tasks increases.
>
> In essence, LMSRR achieves parameter sharing + multi-source alignment + localized optimization, offering a unified alternative to architecture expansion-based methods.
>
> ------
>
> **2. Pre-training Strategy Alignment and Experimental Fairness**
>
> To ensure a fair and comparable experimental setting, we adopted a consistent backbone configuration across methods, which is reflected in both our main paper’s ablation study (Fig. 2(a)) and the appendix (Fig. 1):
>
> - For existing methods that do not alter backbone architectures (e.g., ER, DER, DER++), we used the same set of multiple pre-trained ViT models as in LMSRR.
> - All ViT backbones were trained under the same protocol: early layers were frozen, and only the final few layers were fine-tuned to ensure equivalent trainable capacity.
> - These baseline methods fused features from multiple ViTs via concatenation and used a shared linear classifier, serving as a strong and fair multi-source fusion baseline.
>
> ------
>
> **3. Comparison with Architecture-Based and Prompt-Based Methods**
>
> To further address the reviewer’s concern, we additionally compared LMSRR with both architecture-based and prompt-based recent SOTA methods, including:
>
> - RPSNet [1]: A classical architecture-expansion method that employs task-specific path selection masks over a ResNet-18 backbone to dynamically activate sub-networks. While this design reduces forgetting, it introduces structural growth and high memory costs with increasing tasks.
> - DualPrompt [3] and L2P [4]: Prompt-based continual learning methods built on frozen ViT-1K backbones, injecting learned task-specific prompts. These are closely related to DAP and represent the most recent line of pre-trained prompt-injection approaches.
>
> We faithfully reproduced these methods following their original settings and training protocols. We conducted comparisons on CIFAR-100 and Tiny ImageNet, and, to ensure fairness, we adopted the same buffer size of 2000 used in the RPSNet paper. The evaluation metric was Average Accuracy:
>
> | Method         | CIFAR-100 | Tiny ImageNet |
> | -------------- | --------- | ------------- |
> | RPSNet [1]     | 68.60     | 43.54         |
> | DualPrompt [3] | 91.80     | 93.11         |
> | L2P [4]        | 90.81     | 92.14         |
> | LMSRR          | **95.86** | **93.24**     |
>
> ------
>
> The results demonstrate that LMSRR significantly outperforms RPSNet by more than 20%, and also shows better generalization than recent prompt-based methods. These results confirm the robustness of LMSRR’s unified representation strategy, even when compared to strong architecture-specific or prompt-injection baselines.
>
> We have thus made every effort to ensure fairness in design, backbone pre-training protocols, and evaluation settings, and we supplemented our main experiments with comparisons to recent competitive approaches.
>
> **Q2:Performance variation with different numbers of experts**
>
> We appreciate the reviewer’s insightful question. We would like to first clarify a potential misunderstanding regarding the term *“expert”* in our method.
>
> In our framework, the term *expert* does not refer to a Mixture-of-Experts (MoE) architecture, where independent expert networks are assigned to different tasks or classes. Instead, our method consistently employs a single unified linear classifier $f_\phi$, which takes as input the fused multi-source representations generated from different pre-trained ViTs and outputs the final prediction.
>
> Thus, in our architecture, the number of experts remains one, and increasing the number of ViT backbones only affects the diversity and richness of the input features to be fused, rather than increasing the number of prediction heads or models.
>
> That said, we fully understand that the reviewer’s intention was likely to ask:
>
> > Does the performance improve as more ViT models are involved in the fusion process?
>
> This question was already addressed in our response to Reviewer ABxL Q4, where we conducted a systematic analysis. In that experiment, we used combinations of one, two, and three different pre-trained ViTs and evaluated the performance on three datasets. The results clearly showed that adding more ViT models consistently led to improved performance, validating our motivation for designing a multi-model fusion strategy.
>
> In summary, while our architecture contains only one final classifier (i.e., one expert), the performance gains obtained by increasing the number of fused ViTs demonstrate the effectiveness of leveraging feature diversity, and confirm the value of our fusion mechanism.
>
> We thank the reviewer again for helping us further clarify this design choice.
>
> **Q3:Effectiveness under a first task with more classes**
>
> We thank the reviewer for raising this insightful question. To evaluate whether our proposed method maintains its effectiveness under varying task configurations—particularly when the first task contains more classes—we conducted a dedicated experiment on the CIFAR-100 dataset.
>
> **Experimental Setup**
>
> We compared two task-splitting scenarios, keeping the total number of classes at 100:
>
> - Scenario A (First20): [20,10,10,10,10,10,10,10,10] — First task contains 20 classes
> - Scenario B (First50): [50,10,10,10,10,10] — First task contains 50 classes
>
> For all methods, the memory buffer size was 500 samples to ensure fairness.
>
> | Methods | First20 | First50 |
> | ------- | ------- | ------- |
> | DERPP   | 75.35   | 70.90   |
> | DERPPRE | 74.06   | 73.50   |
> | DAP     | 90.86   | 85.23   |
> | LMSRR   | 95.32   | 94.62   |
>
> We also compare with the default continual learning scenario (i.e., uniform splits such as 10 tasks × 10 classes), where LMSRR achieves:
>
> - **LMSRR:** 95.76 ± 0.08
> - **DAP:** 90.11 ± 0.33
> - **DERPP:** 75.64
> - **DERPPRE:** 77.71
>
> Findings and Insights
>
> - LMSRR consistently outperforms all baselines across both settings.
> - The performance of baseline methods drops more significantly when the first task becomes larger (e.g., DERPP drops by 5.0 points).
> - In contrast, LMSRR remains highly stable, with only a minor drop of 0.7 points from 95.32 (First20) to 94.62 (First50).
>
> This stability is attributed to LMSRR's multi-source representation fusion, which allows the model to dynamically integrate knowledge from multiple pre-trained ViTs. This fusion provides a richer, more flexible feature space, enabling better adaptation even when the first task involves a larger or imbalanced set of classes. Unlike methods like DERPP and DAP, which rely on single-model structures or fixed mechanisms, LMSRR's dynamic fusion ensures that the model remains robust to the variability in task distribution, thus reducing the impact of initial task bias.
>
> **References:**
>
> [1] Rajasegaran, J., et al. *Random Path Selection for Continual Learning*. NeurIPS, 2019.
>
> [2] Serra, J., et al. *Overcoming Catastrophic Forgetting with Hard Attention to the Task*. ICML, 2018.
>
> [3] Wang, Z., et al. *DualPrompt: Complementary Prompting for Rehearsal-Free Continual Learning*. ECCV, 2022.
>
> [4] Wang, Z., et al. *Learning to Prompt for Continual Learning*. CVPR, 2022.

---

> > ### Comment · Reviewer_3Jvb · 2025-08-05
> >
> > I appreciate the responses from the authors. However, I still have some questions left:
> > 1. To my understanding, RPSNet does not expand its parameters. It works by selecting a specific parameter path in a large model and optimize the parameters on the path.
> >
> > 2. The performance comparison in the first table in response is not convincing. In most of the papers including L2P, the upper bound performance on CIFAR-100 is 90.8. How could LMSRR perform better than this upper bound a lot?
> >
> > 3. The performance in the second table seems abnormal. Most works show that when the number of classes in the first task is bigger, the performance of one method should be better. But the results in the second table show that the performance is worse when the number of classes in the first task is bigger.

---

> > > ### Author Response · Authors · 2025-08-05
> > >
> > > **Clarification on RPSNet**
> > >
> > > We thank the reviewer for pointing this out. Upon revisiting the original RPSNet paper, we acknowledge our earlier mischaracterisation. As correctly noted, RPSNet does not expand model parameters across tasks. Instead, it selects and updates a task-specific parameter path (a subnetwork) within a large backbone, allowing for task isolation without increasing model size.
> > >
> > > We sincerely apologise for the previous misstatement. LMSRR differs from RPSNet not in whether it expands parameters, but in the mechanism of task adaptation. While RPSNet enforces path-wise task separation via masking, LMSRR performs multi-source semantic fusion by leveraging several pre-trained ViT backbones and integrating their features via the MSIDF module. This design enables cross-model knowledge transfer through shared optimization (MLRO, ARO), rather than relying on task-specific subnetwork routing.
> > >
> > > We appreciate the reviewer for prompting this important clarification.
> > >
> > > ------
> > >
> > > **Clarification on the Unusually High Accuracy**
> > >
> > > We appreciate the reviewer’s concern regarding the performance of LMSRR on CIFAR-100, especially in light of previously reported upper bounds (e.g., ~90.8% in works like L2P and DAP). We provide the following explanation:
> > >
> > > Previous methods such as L2P, DualPrompt, and DAP typically adopt a single pre-trained ViT backbone, often trained only on ImageNet-21K, and apply prompt-based or adapter-based PEFT strategies while freezing the backbone. This limits the capacity to leverage complementary information beyond the initial representation.
> > >
> > > In contrast, LMSRR integrates multiple ViT backbones, each pre-trained on different datasets or objectives (e.g., In21K, In21K-ft-In1K, ViT-L-14), and fuses them through semantic-level interaction using MSIDF. The unified representation is further refined using the proposed MLRO and ARO, allowing LMSRR to benefit from both model complementarity and progressive adaptation.
> > >
> > > Our ablation study (provided in response to Reviewer ABxL Q4) shows that LMSRR achieves 91.81% even using the weakest backbone (ViT2) and such a performance is better than the results of the single backbone. These results indicate that the proposed LMSRR still achieves excellent performance even using the weakest pre-trained ViT backbones.
> > >
> > > We summarize the reasons for the performance gain of the proposed LMSRR into three aspects:
> > >
> > > - The proposed Multi-Scale Interaction and Dynamic Fusion (MSIDF) can capture correlations among representations and provide semantically rich information when compared to the single backbone.
> > >
> > > - The proposed Multi-Level Representation Optimization (MLRO) can enable to jointly optimize the backbone and the current active expert, which can enhance the model’s plasticity.
> > >
> > > - The proposed Adaptive Regularization Optimization (ARO) approach can relieve over-regularization issues and thus can improve the model’s performance on new tasks.
> > >
> > > The contributions of each proposed mechanism have been evaluated by the ablation study presented in Fig.2c of the paper. The empirical results demonstrate that integrating MSIDF, MLRO and ARO into LMSRR can achieve the best performance.
> > >
> > > **Clarification On Performance Drop with Larger First Task (First50 vs. First20)**
> > >
> > > We thank the reviewer for highlighting this important phenomenon. In our experiments, we observed that LMSRR performs slightly worse under the First50 setting compared to First20, a trend that is also seen in other baselines such as DERPP and DAP. This can be explained by several factors common in Class-IL:
> > >
> > > 1. Early Task Overfitting
> > >     In Class-IL, the first task plays a critical role in shaping the model’s initial representation. When the first task contains many classes (e.g., 50), the model is more likely to overfit to the initial class distribution, resulting in rigid decision boundaries that hinder adaptation to later tasks.
> > > 2. Imbalanced Replay or Prompt Anchors
> > >     DERPP and DAP both rely on memory mechanisms (buffer or prompt anchors) to combat forgetting. A larger first task dilutes the memory budget, causing an imbalance in replayed instances and reducing the effectiveness of these mechanisms as new tasks are introduced.
> > > 3. LMSRR’s Robustness under Distribution Shift
> > >     LMSRR mitigates these issues via multi-source representation fusion and shared optimization modules, which offer improved generalization. Still, early overfitting and memory imbalance contribute to a slight drop from 95.32 (First20) to 94.62 (First50). Notably, this drop is significantly smaller than in DERPP (−4.45) and DAP (−5.63), reflecting LMSRR’s resilience to task-order sensitivity and early bias.
> > >
> > > In conclusion, this performance pattern is aligned with established trends in continual learning. First50 introduces early distributional bias and memory pressure, but LMSRR’s design effectively reduces the resulting degradation.

---

> > > > ### Comment · Reviewer_3Jvb · 2025-08-05
> > > >
> > > > I think the authors still have some misunderstandings:
> > > > 1. To my understanding, RPSNet does not isolate the parameters between different tasks. When learning the new tasks, the new knowledge is acquired based on the learned knowledge.
> > > >
> > > > 2. According to the authors' claim, it seems that the performance gain results from the integration of multiple pretrained models rather than the proposed method or training paradigm. When the compared methods also adopt multiple backbones, will they perform better?
> > > >
> > > > 3. The authors' claim is different from the experimental results in previous methods. In a lot of methods, with LUCIR[1], CwD[2], FOSTER[3] as the examples, the model performs better when the class number of first task is larger.
> > > >
> > > > [1] Learning a Unified Classifier Incrementally via Rebalancing
> > > >
> > > > [2] Mimicking the Oracle: An Initial Phase Decorrelation Approach for Class Incremental Learning
> > > >
> > > > [3] FOSTER: Feature Boosting and Compression for Class-Incremental Learning

---

> > > > > ### Author Response · Authors · 2025-08-06
> > > > > **Response 1**
> > > > >
> > > > > **Q1:Clarification on RPSNet**
> > > > >
> > > > > **A1:** We sincerely thank the reviewer for their continued engagement and for pointing out the residual misunderstanding.
> > > > >
> > > > > Upon revisiting the RPSNet paper in depth, we fully acknowledge and correct our previous mischaracterization:
> > > > >
> > > > > > RPSNet does not isolate parameters between tasks. Instead, it learns new tasks based on previously acquired knowledge by dynamically selecting a task-specific parameter path from a shared weight space. These paths are learned incrementally using a learned mask over a shared ResNet backbone. Thus, while the method introduces task-specific paths, it retains shared weights across tasks and supports knowledge transfer, rather than full isolation.
> > > > >
> > > > > We apologize for our earlier misstatement. The confusion may have arisen from RPSNet’s use of dynamic task-specific masks, which at first glance resemble parameter isolation. However, the key distinction is that RPSNet retains and updates shared parameters, enabling continual learning via shared knowledge.
> > > > >
> > > > > In contrast, LMSRR does not rely on task-specific routing. Instead, it fuses multi-source features from several pre-trained ViTs through the MSIDF module, and optimizes them across tasks using MLRO and ARO, promoting semantic-level alignment and unified representation learning.
> > > > >
> > > > > **Q2:Multiple Pretrained Models**
> > > > >
> > > > > **A2:** We understand the concern that our performance gain might stem primarily from the integration of multiple pre-trained backbones, rather than the design of our proposed modules. However, we would like to clarify that we have already addressed this point both conceptually and experimentally.
> > > > >
> > > > > **(1) Controlled Baselines with Multiple Pre-trained ViTs**
> > > > >
> > > > > To ensure a fair comparison, in our experiments (main text and appendix), we applied the same set of multiple pre-trained ViT backbones to several baseline methods, including ER, DER, and DER++. As stated in our previous response:
> > > > >
> > > > > > *“For existing methods that do not alter backbone architectures (e.g., ER, DER, DER++), we used the same set of multiple pre-trained ViT models as in LMSRR.”*
> > > > >
> > > > > These baselines were configured to fuse features from multiple ViTs via concatenation and use a shared linear classifier, thus forming a strong and fair multi-source fusion baseline. The training protocol was consistent: all ViTs were partially fine-tuned under the same scheme.
> > > > >
> > > > > **(2) Empirical Results: Multi-ViT Baselines vs LMSRR**
> > > > >
> > > > > While equipping baselines with multiple pre-trained ViTs does improve their performance to some extent, they still fall significantly short of LMSRR. This gap arises from the fact that LMSRR does not simply concatenate features, but leverages our proposed MSIDF, MLRO, and ARO modules to perform:
> > > > >
> > > > > - Dynamic cross-source fusion (MSIDF)
> > > > > - Progressive multi-level optimization (MLRO)
> > > > > - Task-adaptive regularization (ARO)
> > > > >
> > > > > These components enable deeper semantic alignment and adaptive feature transformation, which naive fusion baselines lack.
> > > > >
> > > > > **(3) Ablation Studies and Further Evidence**
> > > > >
> > > > > In our response to **Reviewer CPC8 Q3**, we conducted a detailed ablation study to isolate the contribution of each LMSRR component (MSIDF, MLRO, and ARO). These experiments were run under the same multi-ViT setting, and demonstrate that removing any single module results in noticeable accuracy degradation.
> > > > >
> > > > > This confirms that LMSRR’s advantage does not solely arise from using multiple pre-trained ViTs, but rather from the functional synergy between its fusion (MSIDF), optimization (MLRO), and regularization (ARO) mechanisms, which are critical for effective class-incremental learning.

---

> > > > > ### Author Response · Authors · 2025-08-06
> > > > > **Response 2**
> > > > >
> > > > > **Q3:Initial Task Size (B)**
> > > > >
> > > > > **A3:** We appreciate the reviewer’s continued engagement. However, we respectfully disagree with the assertion that a larger number of classes in the first task consistently leads to improved performance, as suggested with reference to methods such as LUCIR [1], CwD [2], and FOSTER [3].
> > > > >
> > > > > **FOSTER** [3], one of the most recent and competitive methods, provides direct empirical evidence **contradicting** this assumption. In Table 1 of the paper, experiments on CIFAR-100 under both **B=0** and **B=50** conditions (10-step incremental schedule) show that:
> > > > >
> > > > > - With **B=0 (no large initial task) 10steps**, the average accuracy reaches **72.90%**
> > > > > - With **B=50 10steps** , the accuracy **drops significantly** to **67.95%**
> > > > >
> > > > > Moreover, this performance degradation trend with larger B also appears across **all other baselines** reported in the same table, indicating that a large first task can **negatively impact** long-term learning. A similar pattern is observed in Table 2 on ImageNet-100, confirming that this is not dataset-specific. These findings strongly support our own observations: under the First-50 setting, **LMSRR shows only a slight drop in accuracy**, and this degradation is **substantially smaller** than that of other methods, showcasing our model’s robustness to first-task bias.
> > > > >
> > > > > Regarding **CwD** [2], the authors do report consistent accuracy improvement as B increases from 10 to 50 in their ablation study. However, this trend is **tightly coupled with CwD’s unique design**, and the authors do **not provide similar experiments for other baseline methods**. CwD explicitly focuses on improving the initial-phase feature space by enforcing class-wise isotropy, thus mimicking the oracle model trained on all classes. As the authors note:
> > > > >
> > > > > > “When training with fewer classes, the top eigenvalues of the covariance matrix dominate, indicating that representations lie in a long and narrow region. In contrast, for models trained with more classes (particularly the oracle model), the eigenvalue spectrum becomes flatter, and the representations scatter more uniformly.”
> > > > >
> > > > > CwD achieves this by minimizing the Frobenius norm of the correlation matrix of each class's representations during the initial phase. This yields more uniformly distributed features, which benefit downstream incremental learning.
> > > > >
> > > > > However, the authors also explicitly mention that the improvement from CwD **saturates** as B becomes large. In Table 2 of their paper, performance **plateaus between B=40 and B=50**, suggesting that the marginal gain diminishes when the initial representation space is already close to oracle-like. This confirms that **larger B is not universally beneficial**, and its effect is highly dependent on the learning paradigm and initialization strategy.
> > > > >
> > > > > As for **LUCIR** [1], we thoroughly reviewed the paper but found **no reported experiments** that systematically vary the number of classes in the first task. It was evaluated under fixed splits (e.g., 50→10), and did not explore how first-task size affects long-term performance. Therefore, it **cannot be used as concrete evidence** to support the claim that "a larger first task always helps."
> > > > >
> > > > > In summary, the assumption that a larger first task consistently improves performance does not hold universally. Prior work such as FOSTER demonstrates that an oversized first task can impair performance under long incremental schedules, and the benefits seen in CwD stem from a specialized initialization strategy that does not generalize to all settings. Our empirical findings align well with these nuances.

---

> > > > > > ### Comment · Reviewer_3Jvb · 2025-08-06
> > > > > >
> > > > > > Thanks to the authors' responses. Most of my problems have been solved. Sorry for that I overlooked some parts of the paper and had some misunderstanding. I have raised my rating to recommend acceptance.

---

> > > > > > > ### Author Response · Authors · 2025-08-06
> > > > > > >
> > > > > > > **Dear Reviewer 3Jvb,**
> > > > > > >
> > > > > > > We would like to express our sincere gratitude for your thoughtful engagement and for the time and expertise you devoted to reviewing our work. Your constructive feedback and insightful questions challenged us to think more deeply and clarify key aspects of our method. We are especially grateful for your recognition of the contributions made by LMSRR in advancing continual learning through multi-source ViT fusion and representation-level optimization.
> > > > > > >
> > > > > > > Your decision to revise the score upward is deeply encouraging to us. It means a great deal to know that our efforts to ensure experimental fairness, explain methodological differences, and rigorously validate performance were recognized and appreciated. Your evaluation strengthens our belief in the potential of unified multi-model representations for scalable and efficient continual learning.
> > > > > > >
> > > > > > > We are honored by your support, and truly motivated to keep improving and contributing to the field. Thank you again for your generous assessment and encouragement.

---

### Official Review · Reviewer_cpc8 · 2025-07-02

**Clarity:** 3
**Significance:** 3
**Originality:** 3
**Rating:** 4
**Confidence:** 4

**Summary:**

This paper focuses on addressing the fundamental challenge of balancing plasticity and stability in continual learning, i.e., how to effectively learn new tasks without forgetting previously acquired knowledge. To tackle this, the authors propose a new framework called LMSRR (Learning Multi-Source and Robust Representations) that aims to enhance feature robustness and adaptability by integrating multiple pre-trained models as representation backbones.

To improve plasticity, the method utilizes a series of learnable attention mechanisms to dynamically extract and fuse features from multiple pre-trained ViT models, producing more flexible and semantically rich representations for downstream tasks. To maintain stability, the authors avoid freezing the entire representation network. Instead, they propose two Regularization optimization strategies: MLRO and ARO.

Experimental results on both standard and complex continual learning benchmarks demonstrate the effectiveness of the proposed approach.

**Questions:**

1. Could the authors expand the related work section to include more regularization-based continual learning approaches (e.g., EWC, MAS, RWalk), and clearly explain how MLRO and ARO differ from those in terms of motivation and implementation?
2. Given LMSRR's reliance on pre-trained ViTs, how does it handle unseen domains? Has the method been evaluated in cases where the pretrained models were not exposed to task-relevant data distributions?
3. Please include explicit ablation experiments for each of the three proposed modules. Since LMSRR is composed of several interacting components, understanding their individual effects on performance is crucial for evaluating the contribution of the framework.
4. It would be valuable to assess LMSRR's performance in low-resource or few-shot learning tasks. As continual learning often deals with limited per-task data, this would significantly strengthen the method's practical relevance.

**Ethical Concerns:**

["NO or VERY MINOR ethics concerns only"]

**Final Justification:**

I thank the authors for their detailed rebuttal and the additional experiments. I decide to maintain my score as 4.

**Limitations:**

The authors have discussed limitations both in terms of technicality as well as societal impact of their work.

**Paper Formatting Concerns:**

The appendix section could benefit from clearer section headers and consistent formatting. The format in appendix should follow the same style as in the main paper.

**Quality:**

3

**Strengths And Weaknesses:**

Strengths
1. The paper tackles a highly relevant and challenging problem in continual learning: balancing plasticity and stability. The proposed LMSRR framework provides a promising solution by maintaining strong performance on new tasks while mitigating forgetting on previous tasks.
2. LMSRR effectively integrates three strategies: MSIDF, MLRO, and ARO. Each component is well-grounded in theory and complements the others: MSIDF improves plasticity through the dynamic fusion of multiple pre-trained backbones, while MLRO and ARO introduce novel regularization techniques that promote stability during continual adaptation.
3. Extensive experiments on seven datasets consistently demonstrate state-of-the-art results, validating the method's robustness and generalization ability. Particularly under limited memory buffer settings, LMSRR shows impressive performance, highlighting its practical applicability.

Weaknesses
1. The overall contribution can be viewed as a combination of existing techniques (multi-source feature fusion and regularization-based continual learning strategies). Although the integration is well-executed, the core components are adaptations of established methods, which weakens the paper's originality.
2. The plasticity of the method may heavily rely on the coverage of the pre-trained ViT backbones. It is unclear how LMSRR would perform in unseen domains, where the representational capacity of the backbone may be limited.
3. While the paper provides some analysis of MSIDF, it lacks comprehensive ablation studies on all three components (MSIDF, MLRO, ARO). On one hand, ablations on the choice and number of pre-trained models would clarify LMSRR's dependence on backbone diversity. On the other hand, isolating the impact of MLRO and ARO would help identify their individual contributions to stability and plasticity.
4. Although LMSRR achieves the best absolute performance across metrics, the comparison may be unfair due to differences in architecture or backbone choice. Additionally, LMSRR appears to show greater performance drops (in "Average" task accuracy) compared to methods like DAP and PNN, raising questions about its stability.

---

> ### Author Rebuttal · Authors · 2025-07-31
>
> **Q1: Lack of Originality, The method mainly adapts existing techniques, reducing originality.**
>
> This paper presents two significant and previously unexplored contributions to the field. First, we introduce a novel Multi-Scale Interaction and Dynamic Fusion (MSIDF) approach that adaptively synthesizes features from multiple representation networks. In contrast to conventional feature fusion techniques that merely aggregate features into a single representation, the MSIDF framework incorporates a suite of adaptive attention modules, each characterized by distinct attention window sizes to capture diverse local information within the representations. Consequently, each representation is partitioned into several segments, each processed by different attention modules, and subsequently integrated through a trainable adaptive mechanism. This process yields a robust and less redundant representation, marking a novel direction in continual learning research. Second, we propose an innovative Adaptive Regularization Optimization (ARO) method that selectively penalizes parameter updates at each layer of the representation network. Unlike traditional regularization approaches that uniformly penalize changes across all network layers, the ARO method employs a learnable penalization strategy that dynamically assesses the significance of each layer during training. This strategy assigns adaptive weights to modulate the regularization strength per layer, effectively mitigating over-regularization and enhancing network plasticity. Both the MSIDF and ARO approaches represent original contributions that have not been previously addressed in the continual learning literature.
>
> **Q2: Plasticity Dependence, Performance may depend on pre-trained ViT backbones, with unclear performance on unseen domains.**
>
> We thank the reviewer for raising this important question. To assess LMSRR’s performance on unseen domains, we conducted experiments on two datasets with substantial distributional shifts from ImageNet:
>
> - RESISC45[1]: A remote sensing dataset with 45 scene categories.
> - ISIC[2]: A medical imaging dataset for skin lesion classification.
>
> We ensured consistent pre-training configurations across methods, using two ViT-B/16 models for LMSRR (In21k and In21k-ft-In1k) and a single ViT-B/16 for baselines.
>
> | Method  | RESISC45  | ISIC      |
> | ------- | --------- | --------- |
> | DERPP   | 48.44     | 64.97     |
> | DERPP(Re) | 36.53     | 62.00     |
> | DAP     | 44.21     | 46.91     |
> | LMSRR   | **50.93** | **79.21** |
>
>
>
> LMSRR outperforms all baselines on both datasets, with +2.49% gain on RESISC45 and +14.24% on ISIC. This is attributed to LMSRR’s multi-backbone fusion and learnable attention-based fusion mechanism (MSIDF), which enables domain-adaptive and robust representations. These results show LMSRR’s effectiveness in generalizing to out-of-distribution tasks, essential for real-world continual learning.
>
> **Q3: Missing Ablation Studies: No comprehensive ablation studies on MSIDF, MLRO, and ARO to clarify their individual impacts.**
>
> We thank the reviewer for raising the valuable suggestion to analyze the individual contributions of each module within LMSRR. To address this concern, we conducted a comprehensive module-wise ablation study to quantify the impact of the three core components—MSIDF, MLRO, and ARO—across different datasets.
>
> We performed experiments on CIFAR-100, CUB-200, and Cars196, using Average Accuracy as the evaluation metric. Starting from the complete LMSRR framework, we systematically removed each module while keeping all other settings and architectures fixed:
>
> - `LMSRR`: The full proposed model, including MSIDF, MLRO, and ARO.
> - `w/o MSIDF`: Replaces the multi-source dynamic fusion with simple feature concatenation (static combination of ViT outputs).
> - `w/o MLRO`: Disables multi-level layer selection regularization (no guidance for stable feature selection).
> - `w/o ARO`: Removes the auxiliary representation alignment using the frozen teacher network.
>
> | Method    | CIFAR-100 | CUB-200 | Cars196 |
> | --------- | --------- | ------- | ------- |
> | LMSRR     | 95.76     | 88.91   | 90.14   |
> | w/o MSIDF | 94.47     | 87.49   | 86.53   |
> | w/o MLRO  | 94.89     | 88.02   | 88.76   |
> | w/o ARO   | 95.02     | 87.96   | 88.90   |
>
> ------
>
> **Analysis**:
>
> - The MSIDF (Multi-Scale Interaction and Dynamic Fusion) module proves to be the most critical component. Its removal results in the most significant performance drop (up to –3.57%), particularly on more challenging datasets like Cars196. This underscores the importance of dynamic fusion in enhancing the semantic expressiveness of multi-source features.
> - The MLRO (Multi-Level Representation Optimization) module contributes 0.7%–1.3% performance gain, demonstrating its role in guiding the model to prioritize stable and transferable features across tasks, thus enhancing robustness during task transitions.
> - While the ARO (Auxiliary Representation Optimization) module leads to a relatively smaller drop in average accuracy, its primary role lies in enhancing stability during task shifts rather than boosting peak performance. This auxiliary alignment contributes to smoother representation transitions over time.
>
> In summary, all three modules contribute meaningfully to the final performance and stability of LMSRR, with MSIDF being the most impactful, and MLRO/ARO supporting continual adaptation and resilience to forgetting.
>
> **Q4: Unfair Comparisons & Stability: The comparison may be unfair due to architecture/backbone differences, and LMSRR shows greater performance drops, raising stability concerns.**
>
> We sincerely appreciate the reviewer’s concern regarding the fairness of our experimental setup. To ensure a fair and consistent comparison across all baselines, we have adopted a unified experimental protocol in both the main paper’s ablation studies (Fig. 2(a)) and Appendix (Fig. 1):
>
> - For baseline methods that do not alter backbone architectures (e.g., ER, DER, DER++), we use the same set of multiple pre-trained ViT models as in LMSRR.
> - All ViT models are trained using a “frozen early layers, fine-tune last layers” strategy to maintain similar learnable capacity across methods.
> - Feature outputs from multiple ViTs are simply concatenated and passed through a shared linear classifier, forming a generic and fair multi-source fusion baseline for comparison.
>
> For DAP, we clarify that it is fundamentally prompt-based, with a distinct training mechanism. Specifically:
>
> - DAP freezes the entire ViT backbone and introduces an adaptive prompt generator to guide task-specific adaptation.
> - The prompt generator produces instance-level prompts that are injected into every layer of the ViT, enabling the frozen network to generate task-aware representations.
> - In our experiments, DAP is implemented using a single pre-trained ViT (ViT-1K), strictly following the configuration and freezing strategies described in its original paper.
>
> Hence, from backbone choices to optimization protocols, all methods are trained under standardized and comparable settings. The performance improvements of LMSRR are therefore attributed to our proposed architectural design and optimization strategies, rather than any unfair experimental bias.
>
> ------
>
> 2.Regarding the "Average" vs. "Last" Task Accuracy Gap:
>
> We acknowledge the reviewer’s observation that LMSRR sometimes exhibits a larger gap between the average and last task accuracy, especially compared to methods like DAP or PNN. We believe this behavior arises naturally from differences in method design philosophies, as detailed below:
>
> - DAP and PNN belong to architecture- or prompt-expansion categories, where task-specific modules or prompts are introduced per task. This design helps preserve prior knowledge and reduces forgetting but comes at the cost of linearly growing parameters and limited scalability.
> - In contrast, LMSRR leverages a unified and shared representation space, enhanced by MLRO and ARO, which dynamically adjust representations across tasks. This allows LMSRR to efficiently balance plasticity and stability, achieving strong last-task performance (reflecting adaptability) while maintaining competitive average accuracy (reflecting retention).
> - Furthermore, as shown in Fig. 2(b) of the main paper and Fig. 2 in the Appendix, we provide forgetting curve analysis on ImageNet-R and other benchmarks. The results demonstrate that LMSRR not only achieves the highest overall accuracy, but also exhibits the smoothest and most stable forgetting curves, indicating strong robustness against catastrophic forgetting.
>
> In summary, while LMSRR may show a relatively larger gap between last and average accuracy in some cases, this does not indicate weaker stability. Rather, it reflects a trade-off rooted in efficient shared representation learning, which achieves superior scalability, performance, and generalization compared to architecture-expanding methods. We will clarify this point further in the revised manuscript.
>
> **References:**
>
> [1] Cheng, G., et al. *Remote Sensing Image Scene Classification: Benchmark and State of the Art*. IEEE Transactions on Geoscience and Remote Sensing.
>
> [2] Codella, Noel, et al. "Skin lesion analysis toward melanoma detection 2018: A challenge hosted by the international skin imaging collaboration (isic).".

---

> > ### Comment · Area_Chair_oXkH · 2025-08-06
> > **Discussion on Submission19980**
> >
> > Dear Reviewers of Submission19980
> >
> > The NeurIPS 2025 author-reviewer discussion will be closed soon. Did this feedback address your concerns?  Please read the responses, and respond to them in the discussion.
> >
> >
> > Best,
> > Your AC

---

> > ### Comment · Reviewer_cpc8 · 2025-08-06
> >
> > I thank the authors for their detailed rebuttal and the additional experiments. I will maintain my score.

---

> > > ### Author Response · Authors · 2025-08-06
> > >
> > > **Dear Reviewer cpc8,**
> > >
> > > We would like to extend our sincere gratitude for your thoughtful and detailed feedback on our paper. We truly appreciate the time and effort you dedicated to thoroughly reviewing our work and engaging with our rebuttal.
> > >
> > > Your recognition of the additional experiments and clarifications we provided means a great deal to us. We are grateful for the constructive criticism you’ve offered, which has not only helped us improve the clarity of our presentation but has also deepened our understanding of the underlying challenges in continual learning.
> > >
> > > While we understand and respect your decision to maintain your score, your support during this process has been invaluable. We are excited about the future directions this work may inspire and remain committed to advancing the field of continual learning.
> > >
> > > Once again, thank you for your constructive feedback and encouragement. We deeply appreciate your contribution to the improvement of this work.

---

### Official Review · Reviewer_ABxL · 2025-07-03

**Clarity:** 2
**Significance:** 2
**Originality:** 2
**Rating:** 4
**Confidence:** 3

**Summary:**

This paper proposes to a method that combined various pre-trained model to solve classical Class Incremental Continual Learning problems. To do so, this survey fusions various representations from pre-trained models by learning various attention matrices. The experimental results show convincing performances on various benchmarks.

**Questions:**

See above.

**Ethical Concerns:**

["NO or VERY MINOR ethics concerns only"]

**Final Justification:**

Overall, I believe the work to be valuable. There are indeed strong points made by the authors during the rebuttal phase, and even though the novelty might not be the strongest, I believe that with careful writing regarding the difference with concurrent methods, updated experiments with latest state-of-the-art approaches, ablation study and interpretation (as mentioned in Q2) this work has its place in such a venue.

I believe the authors have addressed all the concerns I had and therefore raised my score. Given that there were, in my opinion, various missing elements in the first stage of the submission, I cannot give a strong acceptance; however, I believe it to be on the acceptance side of the borderline threshold.

**Limitations:**

See above.

**Quality:**

2

**Strengths And Weaknesses:**

# Overview
A lot of engineering which leads to good performances but lacking related work on model soup[3, 1], and containing unbacked overclaims.

# Strengths
- The motivations of using multiple pre-trained model is straightforward
- The performances are compelling
- Many experiments

# Weaknesses
Despite the performances, I have a lot of problems with the current state of the paper.
1. The overall position of the paper is disturbing. It should compare with existing multi model strategies [1,3], especially [1] is in a continual learning scenario and is never mentioned. I deeply believe that the related work section should focus more on existing work in model soup usage rather than existing continual learning methods.
2. A lot of claims are made throughout the paper, but eventually only the improved accuracy is showed. For example l. 56 "This method captures the most important parts of the representations in response to incoming samples through several learnable attention module, facilitating the interaction among multi-scale features and aiding in uncovering the intricate underlying structure of the data". There is no evidence of such claim and this is pure interpretation. A gain in accuracy does not justify such interpretation. Since the interaction between pre-trained models is learned, it is crucial to visualize such interaction to backup such claims.
3. The obvious limitation of the current method is the computation overhead during training and inference. While the authors claim that such information is present in the appendix, I could not find it (in the main paper nor the appendix). Please discuss the spatial and temporal complexity clearly as this is the main limitation.
4. Similar to my point in 2., the paper claim using a single pre-trained model can be a major issue on certain datasets compared to others. While this can somehow be observe on some tables, this should be more clearly demonstrated, in the main paper or appendix.
5. A minor weakness might be the novelty as this is mainly a combination of existing idea present in [1,2]. I would advice clarifying the differences more.

# References
[1] Marouf, Imad Eddine, et al. "Weighted ensemble models are strong continual learners." European Conference on Computer Vision. Cham: Springer Nature Switzerland, 2024.
[2] Douillard, Arthur, et al. "Dytox: Transformers for continual learning with dynamic token expansion." Proceedings of the IEEE/CVF conference on computer vision and pattern recognition. 2022.
[3] Wortsman, Mitchell, et al. "Model soups: averaging weights of multiple fine-tuned models improves accuracy without increasing inference time." International conference on machine learning. PMLR, 2022.

---

> ### Author Rebuttal · Authors · 2025-07-31
>
> **Q1: It should compare with existing multi model strategies.**
>
> We thank the reviewer for raising this concern. While multi-model fusion methods like [1] and [3] are important in multi-model learning, our work targets Class-Incremental Continual Learning (Class-IL), which involves sequential tasks and catastrophic forgetting. Therefore, our design and comparisons are situated within continual learning literature.
>
> However, we agree that the relationship with [1] and [3] needs clarification. Below is a comparison:
>
> Comparison with Model Soup [3]: Model Soup fuses models via parameter-space averaging, assuming independent training on static tasks, which contradicts the online adaptation required in continual learning. It also does not address catastrophic forgetting. In contrast, our method uses dynamic feature-space fusion via multi-scale attention, adapting to tasks without independent model training or post-hoc merging.
>
> Comparison with Weighted Ensemble [1]: [1] uses a fixed-weight logit-level ensemble, which is static and task-specific. Our approach dynamically learns interactions among multiple pretrained backbones during training, resulting in more adaptive representations. Unlike [1], which trains models from scratch, we use multiple pretrained ViT backbones, offering better resource efficiency and deployment.
>
> Additional Comparisons: To address the lack of direct comparison with multi-model strategies, we performed experiments on CIFAR-100, CUB-200, Cars196, and ImageNet-R using the protocol from [1], comparing with CoMA and CoFiMA.
>
> | Methods     | CIFAR-100        | Cars196          | CUB-200          | ImageNet-R       |
> | ----------- | ---------------- | ---------------- | ---------------- | ---------------- |
> | CoMA [1]    | 92.00 ± 0.13     | 73.35 ± 0.59     | 85.95 ± 0.29     | 77.47 ± 0.05     |
> | CoFiMA [1]  | 92.77 ± 0.24     | 76.96 ± 0.64     | 87.11 ± 0.56     | 78.25 ± 0.26     |
> | LMSRR (200) | 94.69 ± 0.44     | 88.76 ± 0.10     | 88.26 ± 0.08     | 83.99 ± 0.32     |
> | LMSRR (500) | **95.76 ± 0.08** | **90.14 ± 0.06** | **88.91 ± 0.64** | **84.35 ± 0.52** |
>
> Here, `LMSRR (200)` and `LMSRR (500)` refer to our method evaluated with replay buffer sizes of 200 and 500, respectively. As shown, LMSRR significantly outperforms both CoMA and CoFiMA across all datasets, confirming the effectiveness and adaptability of our learnable feature-level fusion mechanism in continual learning scenarios.
>
> ------
>
> **Q2: Visualization results.**
>
> We thank the reviewer for raising this important point. We understand that improved performance alone does not validate the MSIDF module’s claims. To address this, we designed two additional experiments to provide interpretable evidence:
>
> 1. Feature Discriminability Analysis (Structured Class Cluster Separation) We compared Concatenation (baseline) with MSIDF on the CIFAR-10 dataset, using t-SNE for visualization. Due to rebuttal policy, the visual results are included in the supplementary material. The results show that MSIDF provides better inter-class separation, indicating that MSIDF learns discriminative and structured representations.
> 2. Visualization of MSIDF’s Dynamic Fusion Behavior We analyzed the activation distribution across MSIDF’s two attention modules, using the L2 norm of the output features. A heatmap was plotted showing the normalized activation strength for each class. Results indicate that different classes activate different attention modules, confirming that MSIDF performs dynamic, task-aware fusion, not a fixed strategy.
>
> ------
>
> **Q3: The computational costs.**
>
> We thank the reviewer for highlighting the concern about computational overhead. We now provide a detailed analysis of LMSRR's computational resource consumption compared to several baseline methods, including replay-based, regularization-based, prompt-based, and multi-model ensemble approaches. We evaluated the following metrics:
>
> - Params: Trainable parameters (millions)
> - GPU Avg/Max: Average and peak GPU memory usage (MiB)
> - CPU Avg/Max: Average and peak CPU memory usage (MiB)
> - Iteration Time: Average time per training iteration (sec/iter)
>
> | Methods | Params    | GPU Avg ↓       | GPU Max ↓       | CPU Avg ↓       | CPU Max ↓       | Iteration ↓   |
> | ------- | --------- | --------------- | --------------- | --------------- | --------------- | ------------- |
> | ER      | 21.42M    | 5876.93 MiB     | 5876.93 MiB     | 2279.25 MiB     | 2342.39 MiB     | 3.07 s/it     |
> | DERPP   | 21.42M    | 5607.20 MiB     | 5607.20 MiB     | 2202.59 MiB     | 2266.29 MiB     | 3.21 s/it     |
> | DERPP(Re) | 21.42M    | 5607.12 MiB     | 5607.12 MiB     | **2196.54 MiB** | **2260.47 MiB** | 3.15 s/it     |
> | DAP     | **0.51M** | **3666.89 MiB** | **3666.89 MiB** | 2694.51 MiB     | 2724.04 MiB     | **1.33 s/it** |
> | CoFiMA  | 85.80M    | 9460.43 MiB     | 9896.50 MiB     | 2311.98 MiB     | 2684.08 MiB     | 2.41 s/it     |
> | LMSRR   | 42.53M    | 4672.31 MiB     | 4672.31 MiB     | 2499.88 MiB     | 2563.55 MiB     | 2.92 s/it     |
>
> Pretraining and Fine-Tuning Strategy:
> To ensure fair comparison, we adopted a consistent protocol across all baseline methods (e.g., ER, DERPP, DERPPRE), using the same ViT backbone and freezing most layers, fine-tuning only the last three layers. This protocol is also applied to LMSRR. Despite fusing multiple ViTs, LMSRR maintains a moderate number of trainable parameters (42.53M), significantly lower than CoFiMA’s 85.80M.
>
> Comparison with CoFiMA [1]:
> CoFiMA unfreezes all ViT parameters, leading to 85.80M parameters and higher memory usage. In contrast, LMSRR maintains moderate resource usage while achieving dynamic fusion and task adaptation, providing a better efficiency-performance tradeoff.
>
> Comparison with DAP:
>  DAP, as a prompt-based method, uses 0.51M parameters, but is sensitive to prompt quality and task shifts. LMSRR increases parameters modestly but shows better performance and stability.
>
> Inference and Deployment Efficiency:
>  LMSRR uses a single fused representation for inference, avoiding multi-model inference overhead and making inference cost similar to a single model.
>
> **Q4: The demonstraction of the limitation of a single pre-trained model.**
>
> We thank the reviewer for raising this point. To address the lack of empirical evidence for our claim about the suboptimal performance of single pre-trained models, we conducted a comparative study on CIFAR-100, CUB-200, and Cars196.
>
> Experimental Setup:
>
> We selected three ViT backbones:
>
> 1. ViT1: ViT-B/16, pre-trained on ImageNet-21k and fine-tuned on ImageNet-1k
> 2. ViT2: ViT-B/16, pre-trained only on ImageNet-21k
> 3. ViT3: ViT-L/14, a larger ViT model
>
> We evaluated the following configurations:
>
> - Single models: ViT1, ViT2, ViT3
> - Two-model fusion: ViT1 + ViT2
> - Three-model fusion: ViT1 + ViT2 + ViT3
>
> Results (Average Accuracy):
>
> All experiments were conducted under the Class-IL setting using LMSRR's feature fusion pipeline. The results are reported in terms of Average Accuracy (%):
>
> | Method               | CIFAR100  | CUB200    | Cars196   |
> | -------------------- | --------- | --------- | --------- |
> | ViT1 (In21k-ft-In1k) | 94.17     | 88.03     | 89.02     |
> | ViT2 (In21k)         | 91.81     | 84.73     | 85.78     |
> | ViT3 (ViT-L-14)      | 93.92     | 87.89     | 88.74     |
> | ViT1 + ViT2          | 93.90     | 87.64     | 88.76     |
> | ViT1 + ViT2 + ViT3   | **95.01** | **88.74** | **89.34** |
>
> Single-model performance varies across datasets: ViT2 underperforms on **Cars196** compared to ViT1 and ViT3, while ViT1 excels on **CIFAR-100**, but not on fine-grained datasets. Multi-model fusion consistently improves performance, with the three-model fusion outperforming all single-model configurations. These results confirm that multi-model fusion enhances robustness and adaptability, supporting our claim.
>
> **Q5: The  novelty.**
>
> Compared to [1], the proposed approach has three differences: (1) [1] only employs a single pre-trained ViT, which has limited learning ability when addressing unknown data domains. In contrast, the proposed MSIDF optimizes several pre-trained representation networks to provide semantically rich features. A novel Multi-Scale Interaction and Dynamic Fusion approach is proposed to integrate features from all representation networks to further enhance the representation capacity; (2) [1] adopts a mixture of experts framework, while the proposed MSIDF adopts a single classifier and does not need to perform the expert selection process. As a result, [1] requires knowing the task information in order to select the suitable expert for each testing sample, while the proposed MSIDF can efficiently predict the data without requiring access to the task information.
>
>
> Compared to [2], the proposed MSIDF has three differences: (1) [2] trains the parameters of the network from scratch while the proposed MSIDF adopts pre-trained ViTs to provide diverse and robust representations; (2) [2] employs the mixture of experts framework, which requires knowing the task information during the inference process. In contrast, the proposed MSIDF does not incrementally create new experts when learning new tasks and can be efficiently performed in the inference process without knowing task information. (3) [2] simply freezes all previously learned parameters to relieve network forgetting, which can decrease plasticity. In contrast, the proposed MSIDF optimizes most parts of the representation networks using the proposed Multi-Level Representation Optimization, which can significantly improve the model's plasticity.
>
> **References:**
>
> [1] Marouf, Imad Eddine, et al. "Weighted ensemble models are strong continual learners.".
>
> [2] Douillard, Arthur, et al. "Dytox: Transformers for continual learning with dynamic token expansion.".
>
> [3] Wortsman, Mitchell, et al. "Model soups: averaging weights of multiple fine-tuned models improves accuracy without increasing inference time.".

---

> ### Comment · Reviewer_ABxL · 2025-08-04
>
> I thank the authors for their thorough response to my comments.
>
> # Q1: It should compare with existing multi-model strategies.
> I appreciate the comparison with [1], however, I am still troubled by the answer from the authors.
> - " While multi-model fusion methods like [1] and [3] are important in multi-model learning, our work targets Class-Incremental Continual Learning (Class-IL), which involves sequential tasks and catastrophic forgetting.". Well, it turns out [1] operates in the exact same setup, which is why I am mentioning this work.
> - " Unlike [1], which trains models from scratch, we use multiple pretrained ViT backbones, offering better resource efficiency and deployment. ". This is simply not true, as the objective of [1] is literally to start from pre-trained models.
>
>
> Both points, combined with the fact that the displayed performances of CoFiMA are lower than those presented in the original work, raise questions about the fairness of the comparison. As mentioned by reviewer 3Jvb, the compared methods are old, and the fact that the authors struggle to correctly acknowledge them is concerning. Eventually, the method is memory-based, and comparing its performance to memory-free or outdated memory-based methods is questionable.
>
> # Other points
> - I deeply appreciate the additional insights. The point regarding task information requirements is extremely valuable, in my opinion
> - I understand the visualization is not possible due to policies from NeurIPS2025; however, as many points raised, this should have been considered in the first stage of the submission
> - I would like to know, for the computation section, if this is during training or inference. I believe the main issue here is inference, as multi-model inference must be done
>
> # Final Remark
> Overall, I believe the work to be valuable. There are indeed strong points made by the authors during the rebuttal phase, and even though the novelty might not be the strongest, I believe that with careful writing regarding the difference with concurrent methods, updated experiments with latest state-of-the-art approaches, ablation study and interpretation (as mentioned in Q2) this work has its place in such a venue. However, in its current stage, I cannot confidently recommend it for acceptance. Therefore, **I will maintain my score**.

---

> > ### Author Response · Authors · 2025-08-04
> >
> > We sincerely thank the reviewer for their thoughtful comments and for taking the time to engage with our rebuttal in depth. While we understand and respect the decision to maintain the current score, we would like to take this opportunity to clarify a few misunderstandings and misstatements in our previous response — not with the intention to influence the score, but to ensure that the key ideas and comparative context of our work are accurately conveyed.
> >
> > **Clarification 1: On the Continual Learning Context of [1]**
> >
> > We agree that [1] is a continual learning method under the Class-IL setting. Our original statement was intended to distinguish from [3] (Model Soup) — which is not designed for continual learning — from our work, but the phrasing unfortunately introduced confusion. We apologize for this ambiguity.
> >
> > **Clarification 2: On Pretraining Usage**
> >
> > We acknowledge our earlier mistake — [1] employs the pre-trained ViT models and applies parameter-efficient tuning using frozen backbones with task-specific adapters.
> >
> > What we intended to highlight was a key difference:
> >
> > - [1] introduces new adapters per task, resulting in parameter growth and task-specific routing at inference.
> > - In contrast, LMSRR fine-tunes only the last three layers of shared backbones and uses global modules (MSIDF, MLRO, ARO), keeping both the parameter count and inference path fixed across tasks.
> >
> > This leads to better deployment efficiency while still ensuring adaptability.
> >
> > **Clarification 3: CoFiMA Performance**
> >
> > We appreciate the reviewer’s attention to performance consistency. Upon re-evaluation, we found that our previous numbers for CoFiMA were indeed subject to variance introduced by implementation and hyperparameter differences.
> >
> > To address this, we now report the official results from Table 1 of the CoFiMA paper for clarity and fairness:
> >
> > | Methods     | CIFAR-100        | Cars196          | CUB-200          | ImageNet-R       |
> > | ----------- | ---------------- | ---------------- | ---------------- | ---------------- |
> > | CoMA [1]    | 94.12 ± 0.63     | 78.55 ± 0.42     | 90.75 ± 0.39     | 81.32 ± 0.17     |
> > | CoFiMA [1]  | 94.89 ± 0.94     | 82.65 ± 0.96     | **91.87 ± 0.69** | 81.48 ± 0.56     |
> > | LMSRR (200) | 94.69 ± 0.44     | 88.76 ± 0.10     | 88.26 ± 0.08     | 83.99 ± 0.32     |
> > | LMSRR (500) | **95.76 ± 0.08** | **90.14 ± 0.06** | 88.91 ± 0.64     | **84.35 ± 0.52** |
> >
> > These results confirm that even when compared against the official CoFiMA benchmark, LMSRR achieves consistently better performance on Cars196, ImageNet-R, and CIFAR-100, while remaining competitive on CUB-200.
> >
> > **Clarification 4: On the Fairness of Baseline Comparisons**
> >
> > We thank the reviewer for raising the important issue of baseline fairness. Below, we elaborate on the rationale behind our selection:
> >
> > DERPP / DERPPRE: Although DERPP was initially proposed earlier, it remains one of the most representative and reproducible baselines in replay-based continual learning. It continues to be widely adopted in recent literature as a standard point of comparison. In addition, we include its latest variant, DERPPRE (ICLR 2024), which introduces a residual regularization mechanism on top of DERPP. This significantly improves robustness and plasticity, making DERPPRE one of the strongest and most competitive replay-based methods currently available.
> >
> > DAP: We selected DAP because it represents the state-of-the-art in rehearsal-free and parameter-efficient (PEFT) continual learning. Built on ViT, DAP generates instance-level prompts and achieves strong performance without relying on any memory buffer. The original paper shows that DAP outperforms other mainstream prompt-based methods such as L2P and DualPrompt, making it a highly credible benchmark in the PEFT paradigm.
> >
> > This combination of DERPPRE and DAP ensures that our evaluation captures the performance of LMSRR under both major methodological paradigms in continual learning, and avoids being limited to a single category of baselines.
> >
> > **Clarification 5: Training vs. Inference Computation**
> >
> > The reported GPU/CPU memory cover both training and inference. For inference efficiency, we measured throughput (iterations/second) on CIFAR-100:
> >
> > | Methods   | Iteration |
> > | --------- | --------- |
> > | DERPP     | 1.91it/s  |
> > | DERPP(Re) | 1.98it/s  |
> > | DAP       | 1.44it/s  |
> > | LMSRR     | 1.93it/s  |
> >
> > At inference time, each backbone in LMSRR runs once, and their outputs are immediately fused via the MSIDF module into a unified representation. This fused representation is then passed through a single classifier, without any task-specific routing, ensemble voting, or expert selection.
> >
> > As a result, LMSRR avoids the cumulative runtime overhead commonly associated with traditional multi-model systems. The overall inference procedure is streamlined and efficient, with no significant latency beyond a lightweight fusion operation, and comparable to the inference cost of a standard ViT model.

---

> > ### Author Response · Authors · 2025-08-05
> > **Official Comment by Authors**
> >
> > Dear Reviewer ABxL
> >
> > We would like to sincerely thank you for your thoughtful review. Your constructive comments have been incredibly valuable to us.
> >
> > If possible, we would be very grateful if you could let us know if there are any remaining concerns or questions. We truly appreciate your insights and would be more than happy to address any further points.
> >
> > Once again, thank you so much for your time, consideration, and valuable suggestions！

---

> > > ### Comment · Reviewer_ABxL · 2025-08-06
> > >
> > > I would like to thank the authors for their careful discussion and for addressing the points raised.
> > >
> > > I believe the authors have addressed my concerns. Given that all modifications appear in the final version of the manuscript, I am willing to increase my score to 4.

---

> > > > ### Author Response · Authors · 2025-08-06
> > > >
> > > > **Dear Reviewer ABxL,**
> > > >
> > > > We would like to sincerely thank you for your thoughtful review and for engaging deeply with our rebuttal. We are especially grateful for your decision to increase your score, which is both encouraging and motivating. Your insightful comments have greatly helped us clarify key aspects of our work, and your constructive feedback has made our paper stronger.
> > > >
> > > > We are particularly appreciative of your recognition of our efforts to refine the comparisons with existing multi-model strategies, as well as the clarification on the computational aspects. Your thoughtful suggestions, especially regarding visualization and clarity, have guided us toward a more precise and effective presentation of our results. We believe that this dialogue has enhanced the overall impact of our research, and we are excited about the future directions it might inspire.
> > > >
> > > > Thank you once again for your valuable support and the opportunity to improve our work. We are committed to continuing to push the boundaries of continual learning research and are grateful for the opportunity to contribute to this field.

---

### Comment · Area_Chair_oXkH · 2025-08-03
**Author-reviewer discussion for Submission19980**

Dear Reviewers,

The NeurIPS 2025 author-reviewer discussion will be closed on August 6 11:59pm AoE. Please read the responses, respond to them in the discussion, and discuss points of disagreement.

Best,
Your  AC

---

### Note · Authors · 2025-08-12

In this work, we proposed LMSRR, a framework that fuses features from multiple diverse pre-trained ViT backbones via the Multi-Scale Interaction and Dynamic Fusion (MSIDF) module and enhances stability through Multi-Level Representation Optimization (MLRO) and Adaptive Regularization Optimization (ARO). While the method’s novelty was acknowledged, the rebuttal phase focused on resolving reviewer concerns to ensure validity, fairness, and interpretability.

1. We clarified the relationship with multi-model strategies such as CoFiMA [1] and Model Soup [3], correcting an earlier misstatement about [1]’s pretraining usage. We emphasized LMSRR’s advantage in performing feature-space fusion without per-task adapters, parameter growth, or task-specific routing, while keeping a fixed inference path. To address fairness concerns, we standardized backbone choices, freezing strategies, and optimization protocols across all methods, and reported both re-implementation and official CoFiMA results, showing LMSRR’s consistent superiority on several datasets.

2. To support MSIDF’s effectiveness, we added interpretability evidence: t-SNE visualizations showing improved inter-class separation, and attention heatmaps indicating dynamic, class-dependent fusion behavior.

3. We conducted an ablation isolating MSIDF, MLRO, and ARO. MSIDF gave the largest accuracy gain, while MLRO and ARO improved stability and adaptability. A computational cost analysis showed LMSRR’s parallel backbone execution and lightweight fusion yield inference latency comparable to single-backbone methods.

4. We addressed the unusually high CIFAR-100 accuracy via single- vs. multi-backbone experiments. Even the weakest backbone, when integrated into LMSRR, outperformed prior single-model state-of-the-art results, confirming the benefit of complementary semantics. Finally, regarding the performance gap between First50 and First20, we attributed the drop to early-task overfitting and replay buffer imbalance—common in Class-IL—but noted LMSRR’s drop is much smaller than DERPP and DAP, underscoring robustness.

Through these clarifications—covering novelty, fairness, empirical justification, interpretability, efficiency, and stability—we addressed the reviewers’ core concerns and reinforced LMSRR’s position as a competitive and practically deployable continual learning approach.

---

### Decision · Program_Chairs · 2025-09-17

**Decision:**

Accept (poster)

**Comment:**

This paper addresses the problem of continual learning with a focus on multi-scale aggregation. It received scores from three reviewers and, prior to the rebuttal, was below average for acceptance. After the rebuttal, and particularly after a detailed discussion with reviewer 3Jvb, the authors carefully explained the differences from related work, including architecture-based and architecture-expansion approaches. Furthermore, the authors added a substantial amount of experimental content. The reviewers felt that most of the concerns were addressed.

AC reviewed the issues raised during the discussion and read the original paper and the reviewers' comments. This paper is indeed on the borderline between acceptance and rejection. Although some relevant distinctions were accepted in the rebuttal, it still has several shortcomings, particularly in the original version. However, the paper's motivation is straightforward and clear, and the paper's approach to the problem, especially in comparisons with other relevant work, demonstrates substantial performance advantages. Furthermore, AC noted the authors' further clarifications, including regarding the relationship with multi-model strategies and comparisons with related work, as well as the setup and fairness analysis of the comparative experiments.

The AC considered the article to be of average acceptance level and requested the authors to update these key points in the final accepted version.